# ProteoAutoNet: high-throughput co-eluted protein analysis with robotics and machine learning

Mengge Lyu[1,2,3,4], Pingping Hu[2,3,4], Guangmei Zhang[2,3,4], Kunpeng Ma[2,3,4], Xuedong Zhang[2,3,4], Pu Liu[5], Sai Zhang[6], Xiangqing Li[7], Rui Sun [2,3,4] ✉, Yi Chen[2,3,4] ✉ & Tiannan Guo [1,2,3,4] ✉

Co-fractionation mass spectrometry (CF-MS) enables large-scale profiling of endogenous protein-protein interactions, yet CF-MS data generation is of low throughput and therefore predictive models are often limited by the scarcity and limited diversity of high-quality training data. To address this, we present ProteoAutoNet, a robotic experimental platform integrated with a computational workflow for high-throughput CF-MS analysis. This workflow increases the throughput of sample processing from protein complex to peptide by about two times. The integrated machine learning model incorporates targeted data augmentation to expand and diversify reliable protein interaction data, thereby improving model robustness. When applied to three thyroid cell lines, the model predicted 25,173 co-eluted proteins with an AUROC of 0.78. This analysis revealed significantly upregulated proteasome and pre-foldin complexes in the lung metastatic follicular thyroid carcinoma cell line FTC238 compared with the normal thyroid cell line Nthy-ori 3-1. Notably, we identified a protein interaction between TGM2 and HK1 that was significantly upregulated in the papillary thyroid carcinoma cell line TPC-1. ProteoAutoNet provides an improved framework for investigating protein-protein interactions and uncovering interactions.

Proteins participate in most biological processes, executing diverse functions through a complex network of protein-protein interactions (PPIs). These interactions drive the assembly of protein complexes composed of multiple physically associated proteins and enable functional inference within protein interaction networks[1,2]. PPIs are critical for advancing our understanding of biological mechanisms and informing drug discovery[3]. Co-fractionation mass spectrometry (CF-MS) is particularly advantageous for capturing such interactions in

native cellular environments, making it a powerful method for discovering PPIs in a near-physiological context[4–7]. Machine learning has emerged as a powerful method for CF-MS analysis, with software like PrInCE utilizing database-derived annotation to predict protein interactions[8]. In addition, meta-analysis has been developed to explore CF-MS data using random forest based on multi-species datasets[9]. While recent studies have reported multiple CF-MS protocols for co-eluted protein analysis[10–12], the sample preparation remains time-

[1]School of Basic Medical Science, Fudan University, Shanghai, China. [2]Affiliated Hangzhou First People's Hospital, State Key Laboratory of Medical Proteomics, School of Medicine, Westlake University, Hangzhou, Zhejiang Province, China. [3]Westlake Center for Intelligent Proteomics, Westlake Laboratory of Life Sciences and Biomedicine, Hangzhou, Zhejiang Province, China. [4]Research Center for Industries of the Future, School of Life Sciences, Westlake University, Hangzhou, Zhejiang Province, China. [5]Westlake Omics Inc., Hangzhou, Zhejiang Province, China. [6]Institute of Automation, Harbin University of Science and Technology, Harbin, China. [7]College of Medical Information and Artificial Intelligence, Shandong First Medical University, Jinan, China. ✉e-mail: sunrui@westlake.edu.cn; chenyi@westlake.edu.cn; guotiannan@westlake.edu.cn

consuming, and analytical methods suffer from suboptimal performance due to the scarcity and limited diversity of high-quality data annotations[9].

Here we built a robotics-assisted platform for CF-MS sample processing. It takes only three days to complete the sample preparation of 540 fractions. We also refined three commonly used complex databases and built an eXtreme Gradient Boosting (XGBoost) based machine learning model for predicting co-eluted proteins with positive sample augmentation. The Complex Portal aggregates protein complexes from 28 species, including 2152 human complexes[13]. The comprehensive resource of mammalian protein complexes (CORUM) is a manually curated and widely used database, including 3637 human complexes[14], while the hu.MAP integrates over 15,000 proteomic experiments data containing 6965 human protein complexes[15,16]. The three integrated databases yielded 96,635 PPIs. To address the scarcity and limited diversity of known true PPIs and enhance the model's ability to generalize, we applied data augmentation to positive samples by introducing random perturbations. The positive-sample augmentation expanded the data to ~2.7 million co-eluted proteins from TPC-1, FTC238 and Nthy-ori 3-1 cell lines for modeling. We next implemented an XGBoost algorithm for the machine learning model that achieved an AUROC of 0.78 on the internal test set and 0.68 on the independent external validation set derived from FTC133 cell line. The model identified 18,811 high-confidence co-eluted proteins in FTC133.

The ProteoAutoNet workflow integrates robotics-assisted protein complex processing and a machine learning model to identify co-eluted proteins in three thyroid cancer cell lines, and compares differentially expressed PPIs. We performed mass spectrometry analysis using data-independent acquisition (DIA) mode[17]. Employing ProteoAutoNet with a 30-minute LC gradient on a QE-HF mass spectrometer, we identified 25,173 protein interactions across three thyroid cancer cell lines. Notably, upregulated protein interaction networks (PINs) were associated with the proteasome and prefoldin complexes in FTC238. The subunits of them are potential markers for thyroid cancer treatment and prognosis, such as PSMB6. The predicted PPI comprising TGM2 and HK1 was upregulated in TPC-1. We further investigated its structure using AlphaFold3.

Overall, this study shows the effectiveness of robotics and XGBoost-based machine learning for identifying and characterizing PPIs in thyroid cancer, providing a scalable and comprehensive approach for high-throughput PPIs research.

## Results

### A robotics-assisted platform for CF-MS sample preprocessing
We established a robotics-assisted sample preparation platform combining Opentrons and JAKA robots[18,19], which processes protein complex samples into peptide samples for LC-MS/MS (Fig. 1A). Opentrons can process up to eight 96-well plates, automating pipetting for protein denaturation, reduction, and alkylation during the SDC-lysed peptide preparation, with up to four plates handled per run in 50-60 minutes. For digestion, it can process up to six plates in 40 minutes per run. The full process, from fractionated complexes to peptides, takes 16 hours for four plates, about 18 hours for eight plates and about 24 hours for 12 plates. The JAKA robot supports open-source, customized workflows for desalting, with a single arm capable of processing up to four 96-well plates simultaneously. Desalting four to six plates five times takes about four hours with one arm. Using two arms, processing eight to 12 plates takes approximately eight hours. When integrated with Opentrons and manual interventions, the platform processes four to 12 plates from complexes to peptides in two to three days.

The integrated platform processed samples from three thyroid cell lines, each with three biological replicates. Each sample was co-fractionated into 60 fractions. A total of 540 fractions from three cell lines were processed in six 96-well plates over three days (Fig. 1B). To evaluate the sample processing performance of the platform, we analyzed the coverage and reproducibility of protein identification using nine unfractionated mixtures from three cell lines. These unfractionated samples were processed alongside the 540 fractionated samples and served as quality controls (Fig. 1C). The biological replicates of the three cell lines identified 2897 (Nthy-ori 3-1), 2938 (TPC-1) and 2986 (FTC238) shared proteins within each cell line. The Jaccard similarity coefficients for protein identification across replicates within each cell line were all greater than 0.84, exhibiting a high degree of reproducibility of the platform (Supplementary Fig. 1A). The high consistency of protein abundance is shown through a Spearman correlation coefficient of over 0.98. (Fig. 1D).

We further compared the robotics-assisted platform to manual processing using unfractionated TPC-1 samples to validate the performance of the platform further. Compared with manual processing group, the coefficient of variation (CV) of protein counts decreased from 1.08% to 1.01% (Fig. 1E). The number of proteins confirmed the reliability of the robotics-assisted platform (Supplementary Fig. 1B). The mean protein counts were comparable between robotic (3408 ± 34) and manual processing (3404 ± 37). Protein counts in the robotical group showed 25th and 75th percentiles at 3398 and 3428 respectively, compared to 3381 and 3437 for manual processing. We also evaluated reproducibility by calculating the CV of protein abundance. The results showed superior quantitative reproducibility in the robotic processing group, with a lower median CV (20.1% in robotic processing versus 21.6% in manual processing) and a narrower interquartile range (12.6% versus 14.4%) (Fig. 1F).

The robotics-assisted platform of ProteoAutoNet provides a scalable solution for high-throughput CF-MS sample processing, particularly when incorporating offline desalting protocols.

### Database curation and machine learning modeling
We refined three protein complex databases, including CORUM, huMAP and Complex Portal by aligning them with the current human proteome, removing absent proteins and unannotated isoforms based on UniProt FASTA (downloaded in May 2024) (Fig. 2A). We developed an extreme gradient boosting (XGBoost) model to predict protein interactions, using the refined databases as the gold standard.

The CORUM includes 4391 human protein entries, with 3428 complexes retained after UniProt alignment[14]. The hu.MAP focused on high-throughput experimental data contains 9929 reviewed proteins from 6901 complexes post-refinement[15]. We selected the human-specific protein complexes from the Complex Portal[13], leading to 3084 reviewed protein entries from 1830 complexes. Across all databases, 1868 proteins were identified as common complex proteins (Fig. 2B). The hu.MAP contains the most unique proteins at 5910, reflecting the high-throughput affinity purification mass spectrometry (AP-MS) experimental origins of hu.MAP. Complex Portal has the fewest unique entries, with 362 proteins. The hu.MAP database significantly overlaps with CORUM, sharing 3413 proteins, indicating its broad coverage. CORUM and Complex Portal share 2116 proteins, representing 48.19% of CORUM and 68.61% of Complex Portal. Transforming complexes into pairs of PINs showed 7128 interactions common to all databases (Fig. 2B). CORUM and Complex Portal shared 10,533 pairs, representing 18.14% of CORUM and 53.98% of Complex Portal, respectively. hu.MAP has the highest unique pairs (52,488), followed by CORUM (44,403), while Complex Portal only contained 4802 unique pairs (Fig. 2B). This discrepancy likely arises from the loss of complex architecture and protein interdependency when converting protein complexes into protein pairs[20]. It highlights the diverse sources of protein-protein interaction (PPI) data across databases, underscoring the need to integrate multiple PPI databases for comprehensive research[14,21]. Converting the complexes into pairs in PINs will increase the number of interactions and may exacerbate the lack of overlap, despite being the prevailing format in co-elution proteomics studies[8,9].

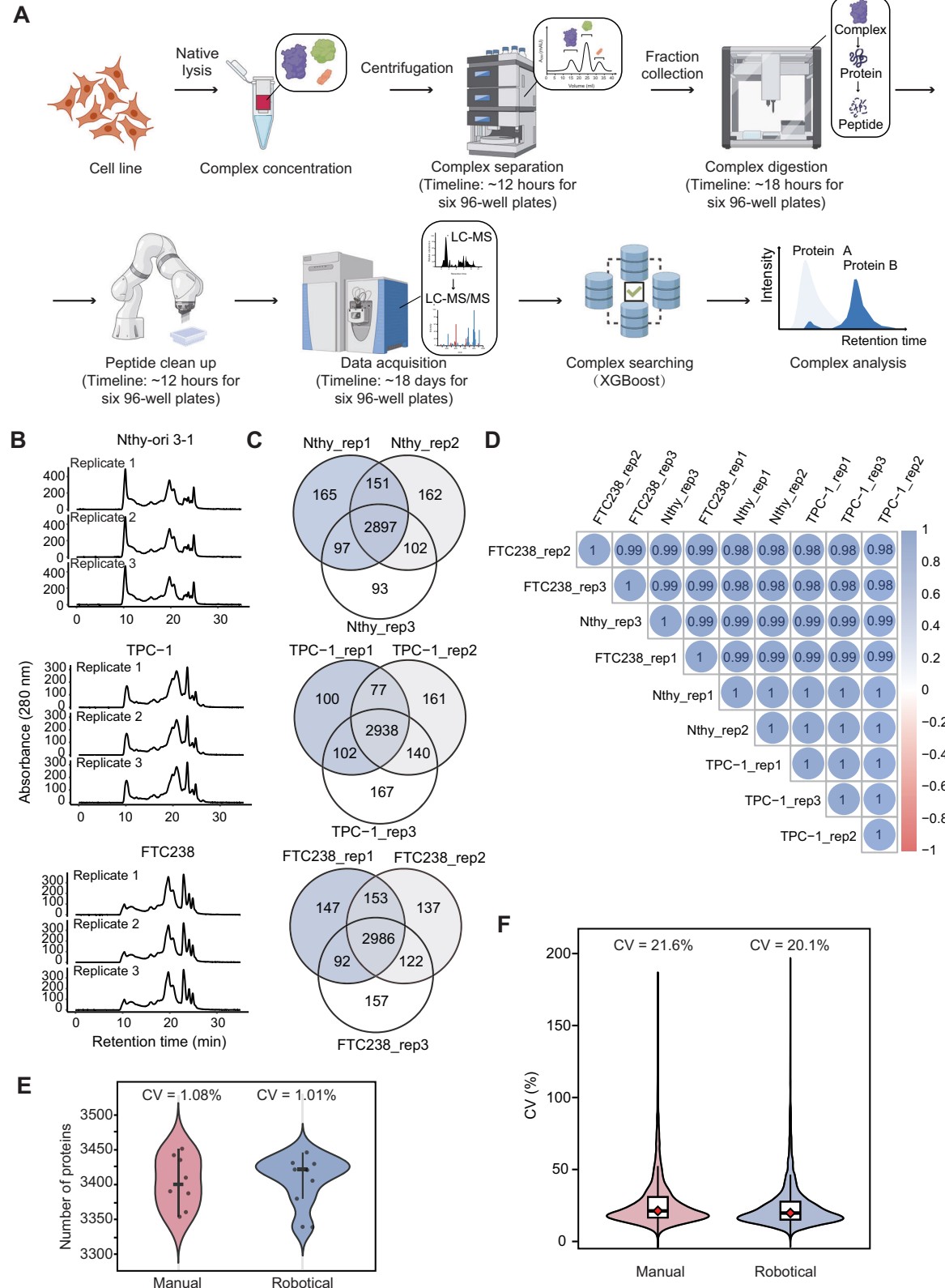

From three integrated databases, we identified 96,635 PPIs with 96,616 being non-monomeric interactions. The combined databases offer a reference for exploring the complexity and dynamics of human PPIs.

Nine protein matrices were annotated using the integrated databases described above, yielding interactions labeled as positive, negative, or unknown (Fig. 2C). A two-stage strategy is implemented to address the scarcity and limited diversity of known true PPIs. The first-

stage filtering employed a 1:5 negative-to-positive ratio to ensure comprehensive coverage of potentially true interactions before downstream augmentation. In the second stage, high-confidence positive interactions (PrInCE score > 0.5) underwent two-tier augmentation. This involved (1) random value perturbation, where protein elution profile values were scaled by a random factor between 90% and 110% and (2) missing-value perturbation, where the positions of

**Fig. 1 | Workflow and performance evaluation of ProteoAutoNet for sample preprocessing. A** Robotics-assisted platform of ProteoAutoNet, including native cell lysis, concentration, fractionation, robotics-assisted peptide generation, LC-DIA-MS analysis, multi-database search of ProteoAutoNet, and differential expression analysis. The time required to process the three cell lines in six 96-well plates is indicated in parentheses. Icons were created in BioRender. Guo, T. (2026) https://BioRender.com/c9p91i3. **B** Size exclusion chromatography traces of 280 nm absorbance from three replicates of three thyroid cancer cell lines. **C** The overlap of protein IDs across three technical replicates by ProteoAutoNet in each cell line. **D** Quality control of the robotics-assisted platform using an unfractionated mixture of cell lysates. Protein abundances were assessed by Spearman correlation after a 30-minute DIA run. **E** Comparative analysis of manual versus robotics-assisted protein identification performance in TPC-1 lysates. The mean number of protein IDs was 3408 ± 34 for robotic processing versus 3404 ± 37 for manual processing (mean ± SD; $n = 8$ technical replicates). The median values (robotical: 3420; manual: 3401) and interquartile ranges (robotical: 3398 to 3428; manual: 3381 to 3437) show the equivalence between sample processing in ProteoAutoNet and manual processing. **F** Comparison of protein abundance variability between manual and robotics-assisted processing in TPC-1 lysates ($n = 8$ technical replicates per group). The analysis included 3567 proteins quantified across all replicates (center line = median; box = Q1-Q3). The robotical group showed lower variability (median CV = 20.1%, IQR = 12.6%, Q1–Q3: 15.5–28.1%) compared to manual group (median CV = 21.6%, IQR = 14.4%, Q1-Q3: 16.9–31.3%).

missing values were randomly either retained in place or shifted to an adjacent time point. Fifty representative traces per original profile were selected from augmented positives based on a composite score. The combined dataset (1:1 negative-to-positive ratios) contained 14.7 to 15 million traces per cell line, split 8:1:1 into training, validation and test sets. The XGBoost model was optimized to maximize the area under the precision-recall curve (AUPRC) via three-fold cross-validation and evaluated performance on internal and external datasets. Our machine learning model enables the prediction of co-eluting proteins in thyroid cancer cell lines, validated in an external dataset.

## A machine learning model for protein interaction prediction

We built an XGBoost model for co-eluted proteins prediction and evaluated its performance using the area under the receiver operating characteristic curve (AUROC) and AUPRC.

In the initial phase, the TPC-1 cell line contained 27,794 positive interactions. The remaining interactions were annotated as either negative (6,139,613) or unlabeled (8,548,311). FTC238 showed comparable positive interactions at 27,615 but exhibited 6,255,162 negative and 8,723,199 unlabeled interactions. Notably, Nthy-ori 3-1 displayed 31,076 positive, 8,397,238 negative and 12,926,198 unlabeled interactions. The greater interaction counts in Nthy-ori 3-1 directly reflect its larger proteome coverage (N), since the potential number of pairs scales with $N^2/2$ (Supplementary Fig. 2A). We evaluated different downsampling ratios and determined that a 1:5 negative-to-positive ratio (precision = 0.28, recall = 0.73) offered the optimal balance, minimizing the excessive false negatives observed at higher ratios. The 1:5 ratio achieved stable precision in random forest models, as confirmed by five-fold cross-validation (Supplementary Fig. 2B). Density plots verified effective enrichment of high-confidence interactions (PrInCE score > 0.5) under this ratio (Supplementary Fig. 2C). We selected 20,635 co-eluted proteins with positive labels from TPC-1, 20,373 from FTC238 and 22,947 from Nthy-ori 3-1.

To address the scarcity and limited diversity of known PPIs, we employed data augmentation by introducing random perturbations to positive samples. The second phase involved data augmentation through random value perturbation and missing-value perturbation as described above (Supplementary Fig. 2D). The XGBoost model was trained using five key features on group-stratified splits of 44.7 million co-elution traces from three cell lines: TPC-1 with 14.7 million traces (14,715,718), FTC238 with 15.0 million (15,005,976), and Nthy-ori 3-1 with 14.9 million (14,969,894). The final 1:1 balanced dataset was divided into training, validation and test sets, with additional FTC133 data for external validation.

Model evaluation showed robust internal performance in cross-validation, with an AUROC of 0.81 (Fig. 3A) and an AUPRC of 0.82 (Fig. 3B). Hyperparameter optimization in the internal validation set yielded an AUROC of 0.77 (Fig. 3C) and an AUPRC of 0.78 (Fig. 3D). Hold-out test results achieved an AUROC of 0.78 (Fig. 3E) and an AUPRC of 0.79 (Fig. 3F). External validation on the FTC133 dataset at the native sample ratio yielded an AUROC of 0.68 (Supplementary Fig. 2G). The reduced AUROC during external validation reflects both technical differences across cell lines and the inherent challenge of class imbalance in native proteomic data. For high-confidence predictions using a strict probability threshold of 0.95, weighted precision achieved 0.5 in the FTC133 dataset (Fig. 3G), which contained 6558 labeled interactions out of 18,811 interactions (precision > 0.5) from integrated databases (Fig. 3H). The three-cell consensus set of 25,173 interactions exhibited comparable performance with a probability higher than 0.95 and a weighted precision over 0.6 (Supplementary Fig. 2E), including 1197 interactions that were previously documented in curated databases (Supplementary Fig. 2F).

Notably, our approach overcomes a critical limitation of existing co-eluted proteins prediction that require multi-dataset integration for reliable performance[9]. Previous single-dataset implementations showed limited predictive accuracy with AUROC of 0.65 as reported[9]. Through systematic integration of data augmentation with the XGBoost algorithm, we achieved a 21.5% improvement in single-dataset predictive accuracy, while maintaining robust external validation performance under native class imbalance.

## Protein interaction landscape of thyroid cell lines

To systematically characterize the biological functions of protein interactions identified across three thyroid cell lines (Nthy-ori 3-1, FTC238 and TPC-1), we implemented a two-tiered analytical framework on high-confidence interactions (weighted precision > 0.6). We firstly conducted Kyoto Encyclopedia of Genes and Genomes (KEGG) pathway enrichment analysis (Supplementary Data 1). Shared interactions among all three cell lines were further analyzed through network visualization and Gene Ontology (GO) term enrichment, revealing conserved functional modules in thyroid biology.

The top ten significant KEGG pathways included ribosome, proteasome, biosynthesis of amino acids, DNA replication and cysteine and methionine metabolism (Fig. 4A). Ribosome and proteasome complexes as the top two pathways align with mass spectrometry-based PPI studies[21,22], further validating the reliability of ProteoAutoNet workflow. We prioritized pathways by combining the p-value and enrichment score. This approach captures stable protein complexes, leading to focus on five major PINs. The proteasome pathway ranked highest and was followed by sulfur metabolism, DNA replication, ribosome, and 2-oxocarboxylic acid metabolism.

ProteoAutoNet detected 37 out of 43 known proteasome components from the KEGG database (Supplementary Fig. 3A), exhibiting the coverage of 20S core particle with its PSMA and PSMB families[23], and the 19S regulatory particle with its PSMC and PSMD families[24]. We also detected partial components of regulatory factors including the PSME and PSMF families. The detected 20S-19S-regulatory factor architecture matches the known proteasome organization[24–26]. This methodological convergence reinforces the validity of ProteoAutoNet for studying the proteasome complex. The sulfur metabolism (7 out of 10 known components) (Supplementary Fig. 3B), DNA replication

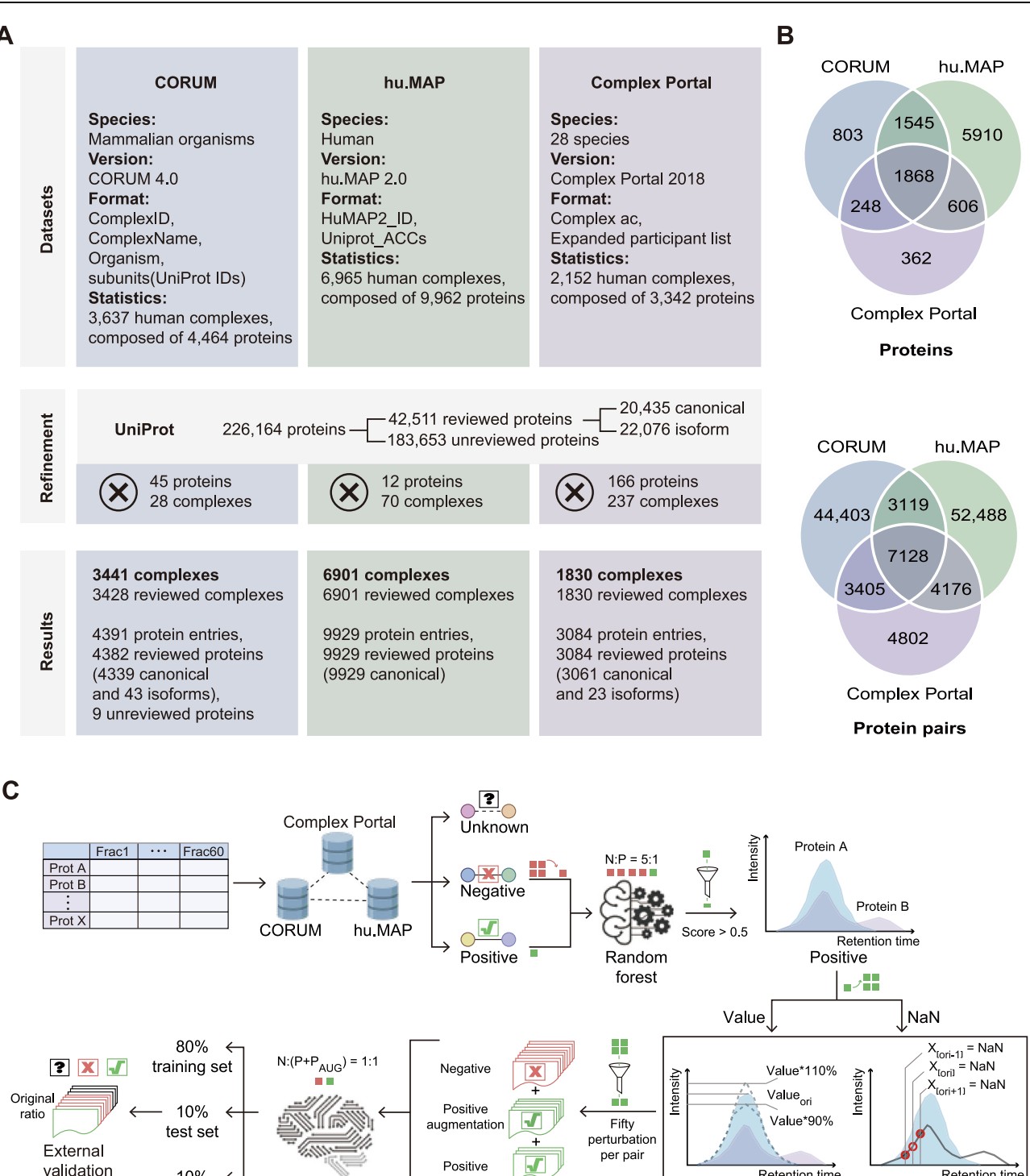

**A**

**Datasets**

**CORUM**

**Species:**
Mammalian organisms
**Version:**
CORUM 4.0
**Format:**
ComplexID,
ComplexName,
Organism,
subunits(UniProt IDs)
**Statistics:**
3,637 human complexes,
composed of 4,464 proteins

**hu.MAP**

**Species:**
Human
**Version:**
hu.MAP 2.0
**Format:**
HuMAP2_ID,
Uniprot_ACCs
**Statistics:**
6,965 human complexes,
composed of 9,962 proteins

**Complex Portal**

**Species:**
28 species
**Version:**
Complex Portal 2018
**Format:**
Complex ac,
Expanded participant list
**Statistics:**
2,152 human complexes,
composed of 3,342 proteins

**Refinement**

UniProt — 226,164 proteins — 42,511 reviewed proteins — 20,435 canonical / 22,076 isoform — 183,653 unreviewed proteins

⊗ 45 proteins / 28 complexes
⊗ 12 proteins / 70 complexes
⊗ 166 proteins / 237 complexes

**Results**

**3441 complexes**
3428 reviewed complexes

4391 protein entries,
4382 reviewed proteins
(4339 canonical
and 43 isoforms),
9 unreviewed proteins

**6901 complexes**
6901 reviewed complexes

9929 protein entries,
9929 reviewed proteins
(9929 canonical)

**1830 complexes**
1830 reviewed complexes

3084 protein entries,
3084 reviewed proteins
(3061 canonical
and 23 isoforms)

**B**

CORUM 803, 1545, hu.MAP 5910, 1868, 248, 606, 362 Complex Portal
**Proteins**

CORUM 44,403, 3119, hu.MAP 52,488, 7128, 3405, 4176, 4802 Complex Portal
**Protein pairs**

**C**

(22/36) (Supplementary Fig. 3C), ribosome pathway (72/131) (Supplementary Figs. 3D, E) and 2-oxocarboxylic acid metabolism (10/18) (Supplementary Fig. 3F) were detected based on KEGG database annotations. We constructed PINs for these pathways and performed Markov clustering (MCL) to identify functional modules.

We displayed common PINs from three cell lines and highlighted edges with experimental evidence in STRING (Supplementary Fig. 3G), and identified three functional modules with extensive overlap with the STRING: translation (Fig. 4B), proteasome-ubiquitin (Fig. 4C) and protein folding (Fig. 4D). As the module with the highest overlap with prior knowledge, the detection of 40S/55S/60S ribosomal subunits and eukaryotic initiation factor 3 (eIF3) complex across all three cell

lines validates the reliability of ProteoAutoNet results (Fig. 4B). Besides, the detection of proteasome corroborates preclinical evidence for proteasome inhibitors in medullary thyroid cancer (MTC)[27]. The Prefoldin and chaperonin-containing TCP-1 (CCT) complexes participate in polypeptide folding, consistent with their canonical roles in eukaryotic protein folding pathways[28]. Some subunits of these complexes have been reported as biomarkers in gastric and lung cancer[29,30].

Although clinical translation remains limited to date, the co-eluted profiles of these complexes and PINs provide a framework to reevaluate their clinical potential. They may guide future studies with improved patient selection.

**Fig. 2 | Comprehensive landscape of protein complex databases and model training workflow. A** Overview of CORUM, hu.MAP and Complex Portal. Databases: summarizing species, database versions, selected columns, and the number of proteins and complexes before refinement. Refinement: detailing FASTA processing and deleted proteins and complexes. Results: presenting the final counts of proteins and complexes. **B** Overlap of proteins across the three databases. Overlap of proteins pairs across the three databases. Protein pairs were derived from the lists of protein complexes. **C** The workflow of the XGBoost model training. Icons were created in BioRender. Guo, T. (2026) https://BioRender.com/bgkoir4. Protein interaction matrices from three cell lines were annotated using three curated databases, with interactions classified as positive/negative/unknown. Positive samples were selected after downsampling negative samples (negative-to-positive ratio = 5:1) in random forest model of PrInCE. High-confidence positives (score >0.5) underwent two-tier augmentation: (i)

value perturbation (90 to 110% random scaling) and (ii) missing-value processing (NA retention or forward/backward shift). For each original pair, 50 perturbed variants were selected by a composite score (Pearson × 30 + valid_points × 0.5 - Euclidean_dist × 0.01). The final dataset comprised ~2.7 million co-elution profiles (balanced 1:1 ratio) partitioned into training (80%), validation (10%), and test (10%) subsets. Model optimization employed area under the precision-recall curve (AUPRC) maximization, with final evaluation on an external validation set ( ~ 200:1 negative-to-positive ratio). Color-coded symbols represent sample categories: green squares for positive samples (with quantity changes indicating augmentation, e.g., 1 square→4 squares), red squares for negative samples (with quantity changes indicating perturbed variants selection, e.g., 4 square→2 squares). The relative quantities of green-to-red squares directly reflect the positive-to-negative sample ratios (e.g. 1:1 ratio shown as 1 green versus 1 red square).

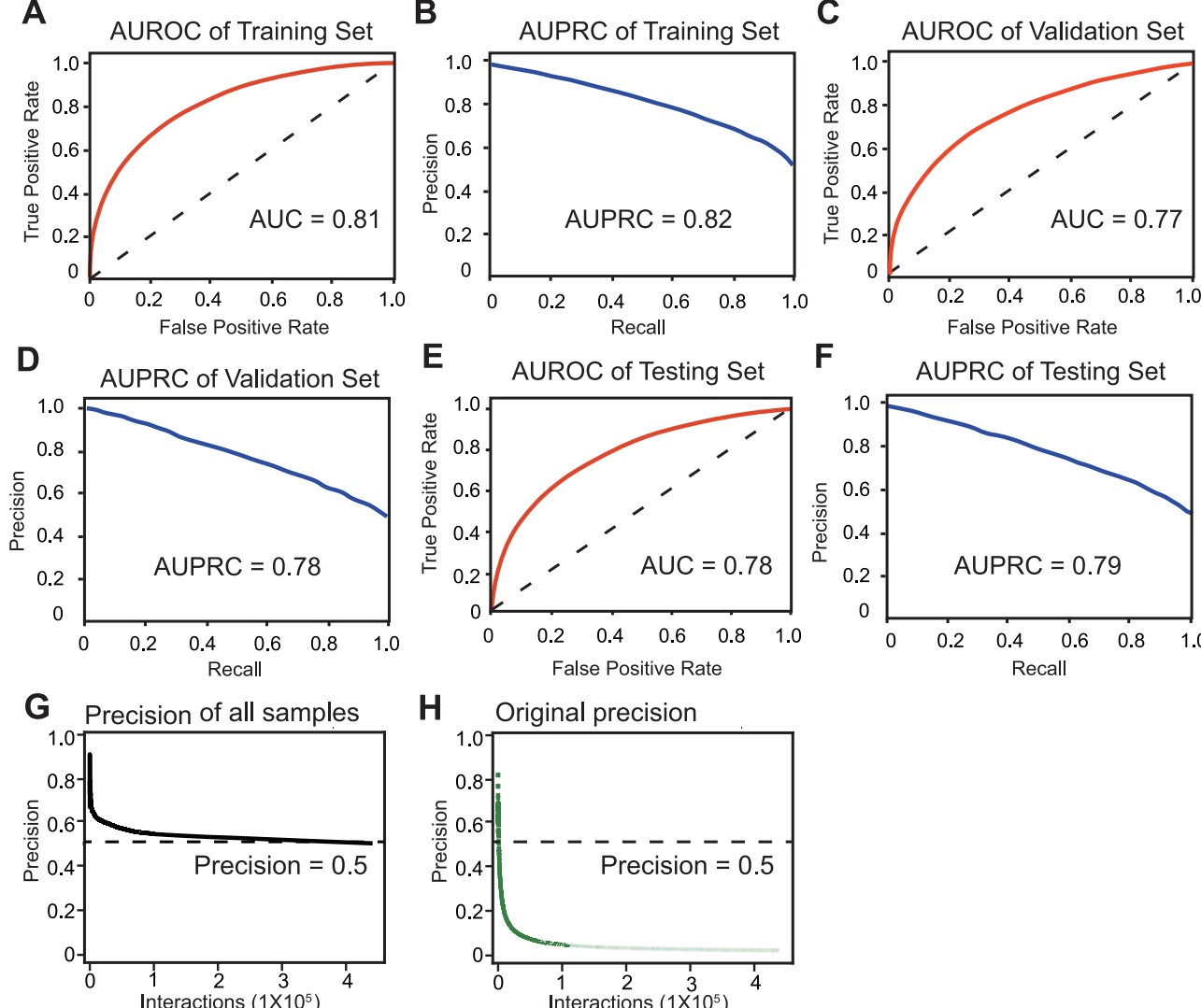

**Fig. 3 | Benchmarking performance of the machine learning model in ProteoAutoNet. A** The receiver operating characteristic (ROC) curve for the XGBoost classifier on the training set (80% of balanced data), showing an area under the curve (AUC) of 0.81. The model was trained on a balanced dataset derived from three cell lines (FTC238, TPC-1, and Nthy-ori 3-1) using a two-phase strategy for positive and negative sample selection. **B** The corresponding precision-recall (PR) curve for the training set, achieves an area under the curve (AUPRC) of 0.82. **C** ROC curve for the validation set (10% of the data), achieving an AUC of 0.77. **D** PR curve for the validation set, with an AUPRC of 0.78. **E** ROC curve for the internal test set (10% of the data), with an AUC of 0.78. **F** PR curve for the internal test set, with an AUPRC of 0.79. **G** Cumulative weighted precision curve for all FTC133 interactions (including positive/negative and unlabeled samples), reaching the precision of 0.5. **H** Precision curve for the labeled samples of the FTC133 dataset.

## Identification of protein interactions and potential biomarkers

Using the normal thyroid cell line Nthy-ori 3-1 as the baseline control, we analyzed PINs in FTC238 and TPC-1 cell lines. PPIs were classified as cancer-associated only when they met the following criteria. The analysis retained PPIs exhibiting retention time shifts (ΔRT ≤ 38 sec) across three replicates, while excluding those proteins with molecular weights smaller than those annotated in UniProt. This dual-filter criterion identified 8984 cancer-related PPIs with reproducible profiles.

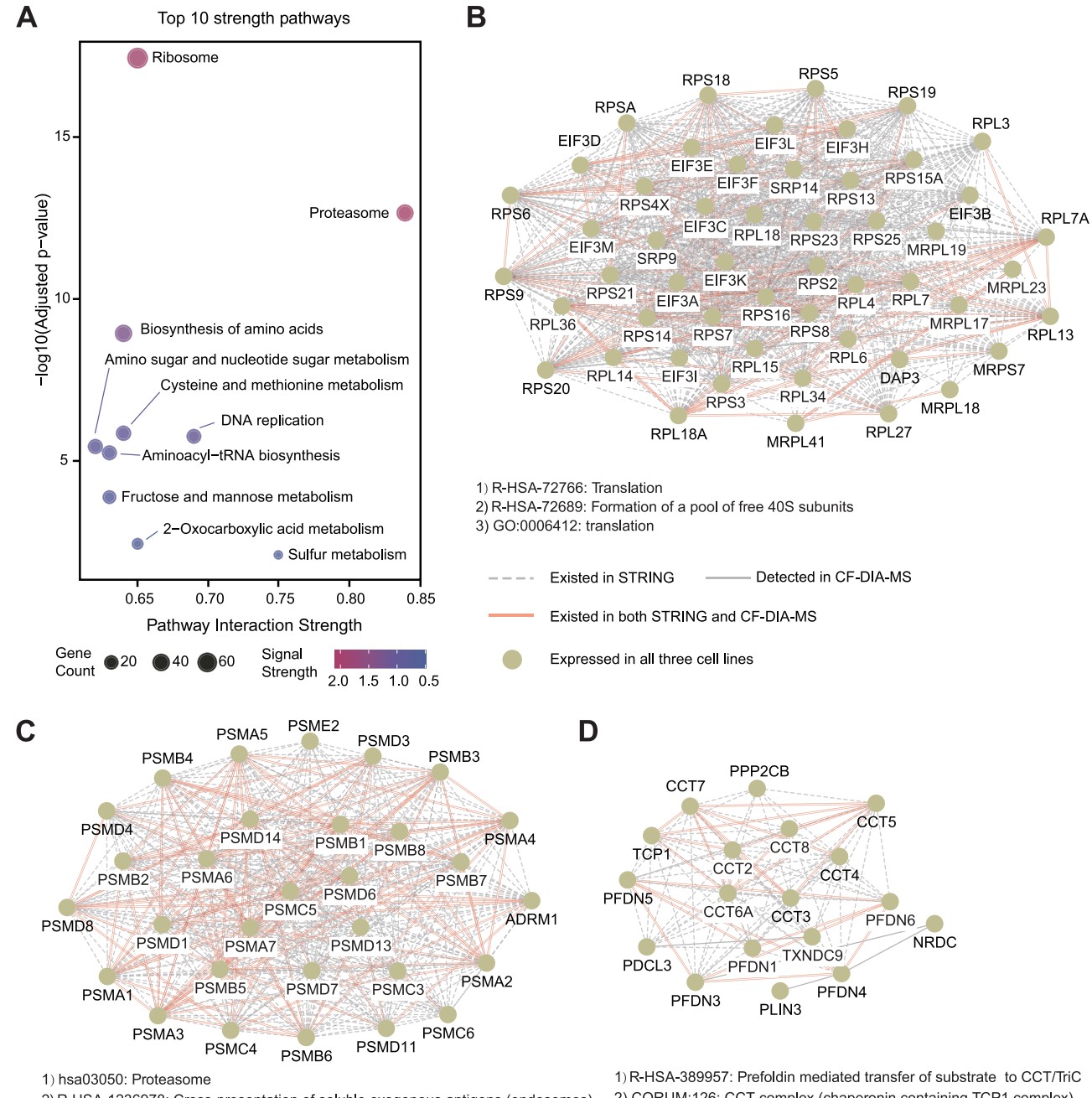

**Fig. 4 | Biological functional analysis of protein interactions from three thyroid cell lines. A** The top ten significantly enriched KEGG pathways. Circle size represents the number of genes enriched in each pathway. Statistical significance was determined using the hypergeometric test (one-sided) in the clusterProfiler R package. P values were adjusted for multiple comparisons using the Benjamini-Hochberg method to control the false discovery rate (FDR).The filling color of the circle indicates the signal strength of the pathway. The horizontal axis indicates pathway interaction strength, and the vertical axis corresponds to the negative logarithm of the adjusted p-value. The names of pathways are annotated adjacent to each point. **B** The protein interaction network of 40S/55S/60S ribosomal subunits and eukaryotic initiation factor 3 (eIF3). The top three enriched pathways are presented beneath the protein interaction network. Nodes represent proteins identified across all three cell lines. Edges: gray dashed lines - interactions in the STRING database, solid gray lines - identified by CF-DIA-MS, red double lines - both CF-DIA-MS and STRING. **C** The protein interaction network of the proteasome-ubiquitin system is shown above, with the top three enriched pathways displayed below it. Node and edge follow the same scheme as in panel B. **D** Sulfur metabolism pathway components were identified, with 7 out of 10 known proteins detected. The enriched pathways are shown in a separate section below the protein interaction network. Node and edge representations follow the same scheme as in panel B.

We identified 2018 downregulated co-eluted proteins and 610 upregulated co-eluted proteins between FTC238 and Nthy-ori 3-1 (Supplementary Fig. 4A, Supplementary Data 2). Similarly, 2198 downregulated co-eluted proteins and 520 upregulated co-eluted proteins were identified between TPC-1 and Nthy-ori 3-1 (Supplementary Fig. 4B, Supplementary Data 2). Differential PPIs from the comparisons of both TPC-1 and FTC-238 cell lines against the Nthy-ori 3-1 were subjected to average-linkage hierarchical clustering (Supplementary Figs. 4C, D). Dendrogram analysis identified functionally distinct modules, with 10 clusters derived from TPC-1 versus

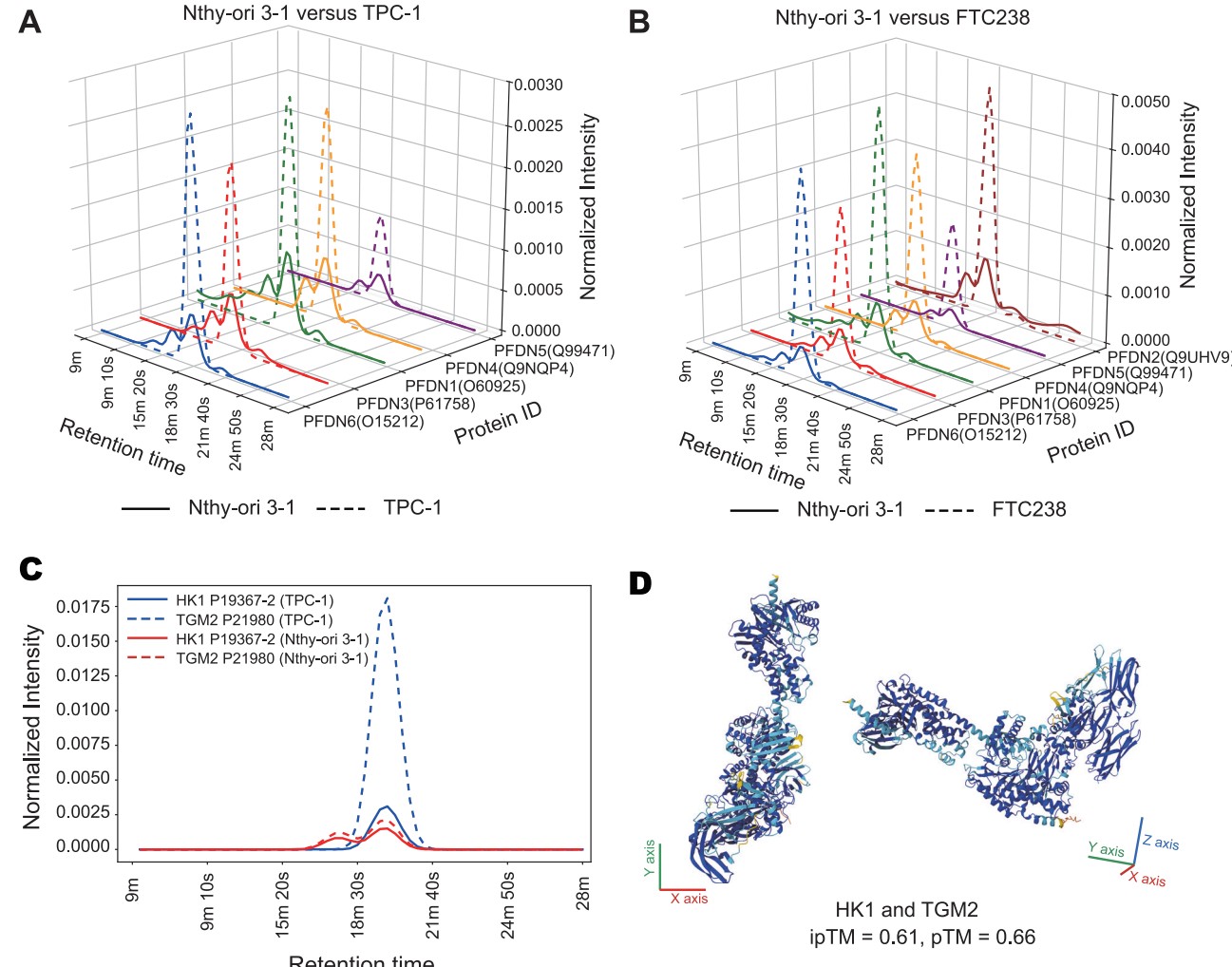

**Fig. 5 | Protein profiles of the Prefoldin complex and predicted interactions with structural modeling. A** Differential protein interactions associated with the Prefoldin complex in TPC-1 versus Nthy-ori 3-1. Curve color denotes protein identity, and line style represents the cell line. Normalized intensity (protein intensity divided by total intensity) averaged across biological replicates. **B** Differential protein interactions associated with the Prefoldin complex in FTC238 versus Nthy-ori 3-1. The scheme of color and line style is the same as panel A. **C** The protein traces of HK1 and TGM2 in Nthy-ori 3-1 and TPC-1. The normalized intensity is protein intensity divided by total protein intensity, followed by averaging across replicates within each cell line. **D** Predicted structural model of HK1 and TGM2 is generated by AlphaFold3. The three axes (X-Y-Z) are shown in the bottom left corner. The scores of ipTM and pTM are 0.61 and 0.66.

Nthy-ori 3-1 PPIs and 9 clusters from FTC238 versus Nthy-ori 3-1 PPIs (Supplementary Data 3, 4). Among them, proteasome and prefoldin complexes showed cancer-related regulation.

Prefoldin is a hexameric molecular chaperone complex associated with various cancers, including bladder, gastric and lung cancers[29–31]. This complex plays critical roles in tumor progression and serves as nano drug delivery actuators due to its structure[32,33]. Notably, PFDN5 splice variants show high expression in malignant thyroid cancer[34] and our study shows upregulated protein levels of PFDN1, PFDN2, PFDN3, PFDN4, PFDN5 and PFDN6 in FTC238 and TPC-1 cell lines (Fig. 5A, B, Supplementary Figs. 5-6). However, global proteomics analyses have not reported prefoldin complex expression changes, potentially due to ubiquitin-proteasome-mediated regulation of PFDN subunits affecting complex stability. This aligns with our observation of elevated proteasome complex expression in FTC238 cell line, consistent with reported proteasome upregulation in lung cancer cell lines[35]. PSMB6 showed significant upregulation in FTC238 (Supplementary Fig. 7), and its overexpression in clear cell renal cell carcinoma is linked to poor prognosis[36]. The proteasome inhibitor bortezomib has shown potential to improve patient outcomes in medullary and anaplastic thyroid carcinoma cell lines[37]. Our study reveals dynamic changes in the prefoldin family

(Supplementary Fig. 6) and proteasome subunits in FTC238, suggesting possible treatment options for metastatic FTC (Supplementary Fig. 7). High MYL9 gene expression is linked to advanced thyroid cancer pathological stages[38]. While MYH10 overexpression is noted in TCGA and head-neck squamous cell carcinoma[39], its interaction with MYL9 remains unexplored (Supplementary Fig. 8).

We identified PINs upregulated in FTC238 and TPC-1 that have not been previously reported. Notably, NRDc interacts with PFDN3, PFDN4, PFDN5 and PFDN6 in FTC238 cell lines, and previous studies indicate that NRDc enhances cell migration response to heparin-binding epidermal growth factor-like growth factor in MDA-MB-453 cell lines[40]. NRDc may regulate cytoskeletal stability and cell migration through prefoldin interactions, particularly with prefoldin subunits (PFDN3 - 6). We also identified another predicted protein interaction comprising HK1 and TGM2 in TPC-1 and Nthy-ori 3-1. Notably, TGM2 exhibited the most marked change within the cluster, and the interaction of TGM2-HK1is significantly absent from STRING (Fig. 5C, Supplementary Fig. 9, Supplementary Fig. 10). HK1 catalyzes the first step of glycolysis, converting glucose to glucose-6-phosphate. Although increased HK1 expression has been reported in PTC[41], but its function in thyroid cancer is not clear. TGM2 is a multifunctional enzyme

involved in cell adhesion and extracellular matrix remodeling, and its overexpression promotes invasion and metastasis of PTC through the activation of NF-κB[42]. The predicted PPI suggests that the metastasis and invasion of PTC may activate metabolic processes, such as glycolysis. The structure of the predicted interactions was further supported by AlphaFold3 (Fig. 5D). Two structural perspectives (X-Y and X-Y-Z) are displayed. The model shows moderate-to-high confidence, with pTM and ipTM scores of 0.66 and 0.61, respectively. It indicates a valuable predicted interaction interface, providing strong structural support for the robustness of this interaction.

The study identifies overexpressed protein interactions in thyroid cell lines, noting dynamic changes in the prefoldin family related to proteasome degradation in FTC238 and TPC-1. This implies that targeting protein interactions could improve biomarker and drug target discovery. Additionally, we identified protein interactions involved in abnormal glycolysis in TPC-1 that were not previously documented, underscoring the utility of ProteoAutoNet for expanding known PINs.

## Discussion

This study introduces ProteoAutoNet, an XGBoost-based machine learning model combined with a robotics-assisted platform, providing an efficient and robust workflow for CF-MS, particularly for the offline clean-up protocol. While recent advancements in CF-MS have introduced automated on-line desalting and peptide digestion, and isobaric labeling has further enhanced sample utilization and reduced manual effort while maintaining data quality[12,43], a semi-automated platform still requires six to 18 days to process 12 to 36 96-well plates for nine biological samples. Here, our sample preparation platform of ProteoAutoNet processes six 96-well plates in three days for nine biological replicates, achieving a twofold throughput improvement compared to previously published platform[12]. The throughput of ProteoAutoNet platform increases non-linearly with sample size, becoming more efficient at larger scales.

The computational component of ProteoAutoNet addresses a fundamental challenge in co-elution-based protein prediction: the inherent scarcity and limited diversity of positive training samples[9]. By combining data augmentation with the XGBoost algorithm, we achieved a 21.5% improvement in single-dataset predictive accuracy, attaining an AUROC of 0.78 in internal testing dataset, while maintaining external validation performance (AUROC = 0.68). The XGBoost-based machine learning model in ProteoAutoNet showed consistent predictive accuracy, with precision exceeding 0.6 for 25,173 interactions across three thyroid cell lines, and predicted 18,811 protein interactions in the independent FTC133 validation set.

We applied the framework of ProteoAutoNet in three thyroid cell lines, processing them from complexes to peptides in three days and generating raw data from 540 fractions within 18 days. The scale of identified proteins was comparable to short gradient CF-DIA/MS runs[43]. We used the QE-HF MS instrument in this study. With better instruments, such as Astral, the proteome depth and analysis throughput could be substantially improved. We successfully identified 25,173 co-eluted proteins and mapped their interaction landscape using ProteoAutoNet. Ribosome, proteasome, and prefoldin complexes emerged as the conserved interactions in thyroid cell lines. Further analysis revealed potential therapeutic targets within the prefoldin family and proteasome complexes in metastatic FTC cell lines, which are challenging to detect in traditional proteomics due to prefoldin degradation via the proteasome pathway. Additionally, we identified a previously unreported interaction between HK1 and TGM2. The activated HK1/TGM2 interaction in TPC-1 cells suggests abnormal metabolic processes in PTC. It indicates an interaction interface confidently predicted by AlphaFold3. Additionally, NRDc/prefoldin subunits indicate that NRDc may regulate cytoskeletal stability and cell migration through its engagement with the prefoldin family in thyroid cancer cell lines.

The stringent filtering criteria for discovering cancer-associated PPIs, which require predicted interactions to exhibit retention time shifts of less than two fractions across three replicates and also exclude proteins with molecular weights smaller than those listed in UniProt, might miss transient and cell-specific interactions. However, this approach successfully revealed previously undetectable interactions involving chaperone proteins and proteasome complexes, providing valuable insights into tumor mechanisms and potential therapeutic strategies. Future research should focus on identifying functional protein interactions by integrating diverse datasets and advancing high-throughput experimental and computational methods for CF-MS.

## Methods

### Cell culture and harvest

The normal human thyroid follicular epithelial cell line Nthy-ori 3-1 (Cellverse Co., Ltd., iCell-h335) was cultured in RPMI 1640 medium (Procell, PM150140) with 10% fetal bovine serum (Hyclone, SV30208.02) and 1% Penicillin-Streptomycin (Hyclone, SV30010). The human papillary thyroid carcinoma cell line TPC-1 (BeNa Culture Collection, BNCC337912) was cultured under the same conditions. The human follicular thyroid carcinoma cell line FTC238 (MeisenCTCC, CTCC-007-0085) has lung metastasis. It was cultured in a 1:1 mixture of Ham's F-12K medium (Procell, PM150910) and high-glucose DMEM (Hyclone, SH30243.01). The culture medium was supplemented with 10% fetal bovine serum (FBS) and 1% Penicillin-Streptomycin solution, matching the concentrations described earlier. The cells were incubated in 5% $CO_2$ at 37 °C. Standard protocols for cell recovery, media replenishment, subculturing, and cryopreservation were strictly adhered to as outlined in the Invitrogen Cell Culture Basics Handbook. Trypsin was added to the culture dish, and the flask was incubated at 37 °C for 5 min. When approximately 80% of the cells were observed to be in a free-floating state, gentle pipetting was used to dislodge the adherent cells from the dish surface. An equal volume of fresh cell culture medium was added to neutralize the trypsin activity. The contents of the culture flask were then transferred to a 15 mL conical centrifuge tube. The cells were centrifuged at 200 g for 3 min, and the supernatant was carefully removed. The cell pellet was resuspended in 10 mL of PBS and centrifuged again at 200 g for 3 min. This washing step was repeated three times with PBS. The cell pellet was resuspended in 10 mL of PBS, and cell counting was performed before stored the cell pellets at -80 °C.

### Native complex extraction and fractionation

The washed cell pellets were resuspended in ice-cold native lysis buffer (pH 7.2) containing 50 mM KCl, 50 mM sodium acetate, protease inhibitors (Roche, cat. 04693116001), and phosphatase inhibitors (Roche, cat. 4906837001). The buffer was added at a ratio of 200 μL buffer per $1 \times 10^7$ cells. The suspension was pipetted up and down ten times using a 1 mL pipette tip and subsequently frozen at -80 °C for 10 min. After thawing on ice, the lysates were centrifuged at 18,000 g for 30 min at 4 °C. The supernatant was transferred to a 30 kDa molecular weight cutoff centrifugal filter and concentrated at 15,000 g for 1 hour at 4 °C. Protein concentration was determined using a BCA assay, and the protein concentration was adjusted to 20 μg/μL with native lysis buffer. The concentrated proteins were centrifuged again at 15,000 g for 30 minutes at 4 °C to remove impurities, and the resulting protein complexes were transferred to sample vials for SEC.

A total of 1 mg of concentrated complexes was loaded onto a chromatographic system comprising two guard columns and a 300 × 7.8 mm BioSep 5 μm SEC-s4000 500 Å (Phenomenex, Torrance, CA). The columns were equilibrated with SEC mobile phase (1× PBS) to 1.5× column volume. The fractionation was performed using an Ultimate 3000 HPLC system. The temperature was set at 5 °C. The flow rate of the SEC column was maintained at 0.5 mL/min for 30 min. Fractions were collected every 19 s from 9 to 28 min and 60 fractions

were obtained totally. Before sample fractionation, a blank run was conducted to verify column cleanliness. A protein standard mix (cat. 69385, Supelco) was used to confirm molecular weight calibration. The blank and standard protein runs were repeated every three sample injections to ensure retention time accuracy. There were 540 fractions generated from SEC column in total.

## Automated desalting and digestion of co-fractionated proteins

The 96-well plates containing SEC fractions were placed in the Opentrons, which is capable of processing four to six plates simultaneously. Sequentially, 30 μL of 5% SDC, 35 μL of 50 mM TCEP, and 35 μL of 200 mM IAA were added to each well in Opentrons. The plates were heated at 100 °C in a water bath for 20 min and cooled to room temperature. Trypsin (3 μL of 0.1 μg/μL) was added, and digestion proceeded for 12 hours. Following digestion, the subsequent steps were performed manually, the solution was transferred to microcentrifuge tubes, 30 μL of 10% TFA was added to terminate the reaction. SDC particles were removed by centrifuging at 18,000 g for 30 min. This step was repeated to ensure peptide recovery. Desalting was performed using JAKA robotics platforms with 96-well desalting columns, simultaneously handling six plates. The desalting columns were first washed with methanol, followed by equilibration with 80% acetonitrile (ACN) containing 0.1% trifluoroacetic acid (TFA). Peptide samples were loaded onto the columns and subsequently washed five times with 2% ACN / 0.1% TFA to remove salts and contaminants. Elution was performed with 40% ACN / 0.1% TFA[44]. Finally, the desalted peptides were dried using a SpeedVac and stored for downstream analyses.

To validate the performance of the automated platform, we compared the robotics-assisted processing against manual processing using eight unfractionated TPC-1 samples for each group. Both sets of samples underwent identical processing conditions following the above protocol for denaturation, reduction, alkylation, digestion, and desalting steps.

Furthermore, to evaluate the sample processing performance during large-scale runs, we analyzed nine unfractionated mixtures from three cell lines that were processed alongside the 540 fractionated samples as quality controls. These quality control samples enabled us to monitor the coverage and reproducibility of protein identification throughout the automated platform operation.

## DIA-MS analysis of the cell lines

Peptides from fractional samples were separated using an Ultimate 3000 nanoLC system (Thermo Fisher Scientific) at 300 nL/min with an effective 30-min gradient. Mobile phase A consisted of 0.1% (v/v) formic acid in water, and mobile phase B consisted of 0.1% (v/v) formic acid in 80% acetonitrile. The column was equilibrated at 7% buffer B, and peptides were eluted with a 30-min effective gradient increasing from 7% to 28% buffer B over 30 min (from 4 to 34 min)[44]. Eluted peptides were ionized and analyzed using a Q-Exactive HF mass spectrometer (Thermo Fisher Scientific) operated in data-independent acquisition (DIA) mode. A full MS scan was acquired over m/z 390–1010 at a resolution of 60,000, with an automatic gain control (AGC) target of 3e6 and a maximum injection time of 80 ms[44]. After the full MS scan, 24 MS/MS scans were acquired. Each MS/MS scan was acquired at a resolution of 30,000, with an AGC target of 1e6 and a maximum injection time set to auto.

We acquired a total of 576 analyzable DIA files, including 540 co-fractionated samples from three cell lines with three SEC replicates (60 SEC fractions per replicate collected every 19 seconds between 9-28 minutes), 27 unfractionated quality control samples of automated platform (three technical replicates per SEC replicate), and 9 batch control injections of pooled three cell line mixtures for MS acquisition. DIA raw files were analyzed using DIA-NN 1.8 (default settings) and against our previously released thyroid cell line spectral library[45]. The cysteine carbamidomethylation was set as a fixed modification, while the methionine oxidation was as a variable modification. The match between run is on. Peptide length range, precursor m/z range, and fragment ion m/z range were set as default. 5% false discovery rate (FDR) of the precursor and peptide and protein was applied. Protein inference and cross-run normalization are off, and quantification strategy is any LC (high accuracy). Other parameters were used by default. The protein matrix that we used for downstream analyses was generated as outlined above.

## Machine learning framework for predicting PPIs

We established a machine learning-based framework for predicting PPIs by integrating three reference databases as the gold standard and employing both random forest and XGBoost algorithms. We extracted the column names of databases using R (v4.3.0) and refined using the human protein sequence FASTA file (downloaded in May 2024) to remove non-matching proteins and complexes, as detailed in Fig. 2A. Complexes from CORUM, Complex Portal and hu.MAP were consolidated and refined through the UniProt website to generate a human interaction reference dataset. Protein interactions were derived from the set of identified proteins (N), yielding a total of N × (N - 1) / 2 possible interactions. However, the set of known true interactions from gold-standard databases is sparse relative to this theoretical maximum.

We implemented a two-phase strategy to build a robust predictive model from these sparse annotations. In the initial phase, sensitivity-optimized downsampling (1:5 negative-to-positive ratio) was performed across all three cell lines (TPC-1, FTC238, and Nthy-ori 3-1) using the PrInCE R package (v1.16.0, R4.3.0). Protein interactions were annotated against integrated database from CORUM, Complex Portal and hu.MAP, with random forest classification (5-fold cross validation) identifying high-confidence interactions (score > 0.5) for data augmentation. The second phase involved two-track data augmentation: (1) random value perturbation, where protein elution profile values were scaled by a random factor between 90% and 110%; and (2) missing-value perturbation, where the positions of missing values were randomly either retained in place or shifted to an adjacent time point. This augmentation process was implemented in Python 3.9.21. Representative augmented pairs were selected based on a composite score: score = (n_valid × 0.5) + (|Pearson_r| × 30) − (euc_dist × 0.01), from which the top 40 and bottom 10 scoring pairs were retained. Final training sets combined original and augmented positives and matched negatives (1:1 ratio). The XGBoost classifier (Python 3.9.21) was trained using stratified partitioning via GroupShuffleSplit() by protein ID to maintain group integrity (80% training, 10% validation, and 10% test sets). Five interaction features were employed: smoothed and raw Pearson correlation coefficients, Euclidean distance, co-peak frequency, and weighted cross-correlation (WCC) scores. Hyperparameter optimization was conducted through three-fold stratified grid search covering max_depth (4-8), learning_rate (0.01-0.1), subsample ratio (0.6-1.0), with early stopping after 20 rounds of no improvement. Class imbalance was addressed by setting scale_pos_weight() to the negative-to-positive ratio. The final model was selected based on AUPRC performance during cross-validation, with optimal parameters retrained on the complete 80% training dataset. Final model performance was evaluated on 10% internal test set and an independent external validation set of FTC133 cell line.

We employed two methods for precision calculation. The first method treated all unknown samples as negative instances (0), and precision was calculated as TP / (TP + FP), where TP denotes true positives and FP denotes false positives. However, this approach may underestimate the true precision, since unknown samples are likely to contain some true positives. Therefore, we also applied a weighted precision method, which assigns the predicted probability of each unknown sample as its positive-class weight. These weighted precisions were combined with labeled positive and negative samples, and

all interactions were ranked by confidence to compute a cumulative precision curve.

## Functional enrichment and network visualization

The PPIs (weighted precision > 0.6) from TPC-1, FTC238 and Nthy-ori 3-1 cell lines were used for functional enrichment analysis and network visualization. Proteins were converted to KEGG gene IDs and KEGG pathway enrichment was performed using R package clusterProfiler (v4.12.6) enrichKEGG() with parameters: gene = converted IDs, keyType = 'kegg', organism = 'hsa', pAdjustMethod = 'fdr', qvalueCutoff = 0.05. The results were sorted in descending order based on pathway interaction strength, and the top ten most significantly enriched pathways were selected for visualization. A scatter plot was generated using R package ggplot2 (v 3.5.1) to display the top ten significantly enriched KEGG pathways. The pathway names were labeled using the geom_text_repel() function from the ggrepel package to prevent text overlapping.

The interactions from three cell lines were integrated and submitted to the STRING website (https://string-db.org). Experimental interactions were retained in STRING, and the resulting edge list was exported for downstream analysis. The global PINs were reconstructed using Cytoscape software (v3.10.3, based on Java 17.0.5). For the top ten KEGG pathways ranked by strength, the corresponding interactions were subjected to functional module identification using the Markov Clustering algorithm (MCL), implemented in the cluster-Maker2 (v2.3.4) of Cytoscape. The PINs were clustered using default parameters to generate clustered sub-PINs for each pathway.

Gene Ontology (GO) enrichment analysis was conducted using the Metascape online platform (https://metascape.org). The gene list derived from Cytoscape was imported into the platform with the organism specified as *Homo sapiens*. The express analysis pipeline was employed to identify enriched GO terms across the three ontology domains: biological process, molecular function, and cellular component.

## Differential expression analysis of PPIs

Protein chromatograms were reconstructed using Python 3.9.21, with protein abundance represented by the area under the chromatographic peaks. The intensity of a protein pair was identified as the sum of the peak areas of the two proteins. For differential analysis of protein interaction pairs, the average abundance across biological replicates was calculated for identified interactions. Likelihood ratio tests were performed with a p-value threshold of 0.05, followed by Benjamini-Hochberg (BH) correction to obtain adjusted p-values of 0.05. Co-eluted proteins with a fold change greater than 1.5 in absolute value, and median fold changes were calculated to determine differential expression levels.

Protein modules in each cell line were defined by cutting the dendrogram at h = 50, producing X primary branches. Assemblies maintaining stable interactions beyond this threshold were classified as module X + 1 for functional characterization. We focuse on the overlapped modules between FTC238 versus Nthy-ori 3-1 and TPC-1 versus Nthy-ori 3-1. The interactions were subsequently compared with the STRING database to investigate their positions and roles within protein interaction networks. Finally, the PINs were constructed from differentially expressed PPIs using the STRING and Cytoscape. The network type was set to 'physical subnetworks' to specifically retrieve interactions indicating physical complex membership, with interaction sources limited to experimentally validated evidence in STRING. For rewiring the differentially expressed interactions, the intensity of each protein was first normalized by dividing it by the sum of all protein intensities at the time point. These normalized values were averaged across replicates within each cell line for display on the protein interaction networks.

The heatmap was displayed using the pheatmap R package (v1.0.13). The protein matrix was averaged within each cell line and row-scaled. Unsupervised hierarchical clustering of rows was conducted using Euclidean distance and the Ward.D2 linkage method. The columns were not clustered. Both row and column names were displayed.

The predicted protein-protein interaction identified through the above analyses was further validated in silico for their structural plausibility using AlphaFold3, which predicts the three-dimensional structure of protein complexes and provides ipTM and pTM scores for the predicted result.

## Reporting summary

Further information on research design is available in the Nature Portfolio Reporting Summary linked to this article.

## Data availability

The proteomics data, the spectral library, and sample information generated in this study have been deposited in PRIDE under the accession number PXD059608. The remaining data are available within the Article, Supplementary Data files or Source Data file. Source data are providd with this paper. Source data are provided with this paper.

## Code availability

The computational framework ProteoAutoNet, including R and Python scripts and visualized files, is publicly available at: https://github.com/guomics-lab/ProteoAutoNet. The code and example files that were used in this study are publicly available on Zenodo at https://doi.org/10.5281/zenodo.18217457.

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

## Acknowledgements

This work is supported by grants from Joint Funds of the National Natural Science Foundation of China (No. U24A20476) to T.G., National Key R&D Program of China (No. 2021YFA1301600) to T.G., National Natural Science Foundation of China (Young Scientist Fund, No. 32401239) to R.S., Zhejiang Provincial Natural Science Foundation of China (LQ24C050002) to R.S., the State Key Laboratory of Medical Proteomics (SKLP-Y202403) to R.S. and Key Technology Research and Development Program of Shandong Province (2022CXGC020510) to X.L. We thank Westlake University Supercomputer Center for assistance in data generation and storage, and the Mass Spectrometry & Metabolomics Core Facility at the Center for Biomedical Research Core Facilities of Westlake University for sample analysis. We thank Prof. Ruedi Aebersold, Dr. Moritz Heusel and Dr. Chen Li for helpful discussions, and Prof. Leonard Foster and Jenny Moon for advice on sample preparation. We also thank Liang Chen for helping with figures.

## Author contributions

T.G., M.L., Y.C., and R.S. conceived and designed the project. M.L., G.Z. and X.Z. performed the cell culture and harvest. M.L., X.L., and S.Z. set up the robotics-assisted platform. M.L., P.H., and K.M. did fractionation and set the robotic platform of CF-MS. M.L., Y.C., and P.L. analyzed the data. M.L. and R.S. created and designed of the figures. M.L., R.S., and T.G. co-wrote the manuscript. M.L., R.S., and T.G. revised the manuscript. T.G. supported the study. All authors reviewed and edited the manuscript.

## Competing interests

T. G. is a shareholder of Westlake Omics Inc. P.L. is a staff of Westlake Omics Inc. The remaining authors declare no competing interests.
