## [Transparent Peer Review file · Nature Communications]

ProteoAutoNet: high-throughput co-eluted protein analysis with robotics and machine learning

Corresponding Author: Professor Tiannan Guo

Version 0:

Reviewer comments:

Reviewer #1

(Remarks to the Author)

Review of NCOMMS-25-11440, "ProteoAutoNet: high-throughput co-eluted protein analysis with robotics and a multi-database search strategy" by Guo and colleagues.

This is an interesting automation workflow with potential broad applications for CFMS style interactomic studies. Overall, the ProteoAutoNet offers the potential as a method for reducing intrasample variability and overall analysis time. However, there are some performance gaps warranting a more thorough comparison between robotically and manually-prepared samples would help to elucidate the relative advantages, and whether there are drawbacks in terms of coverage and replicability. Additionally, integration of the software component isn't fully convincing, as iRefIndex, while well benchmarked, has low overlap with other public databases. Notably, there is approximately a 20% drop in IDs when only considering non-iRefIndex entries, which is worth exploring to improve confidence in the computational aspects of the pipeline.

Specific points warranting attention:

-Use of 9 unfractionated samples: protein abundance is used as QC measure, but overlap of protein IDs and coverage between replicates as well as comparison to non-robotically prepped samples would provide more objective QC measures

-In general, comparison of robotically-prepped samples to standard workflow would give more robust evidence of benefits of robotic workflow outside of time/effort savings -Page 4, line 151, authors state that they use 81 raw LC-DIA/MS files to generate their database. A further explanation of these files, species, fractionation, quantity etc. would help elucidate this database. The methods page lists 576 DIA files.

-iRefIndex identified many more PINs in Figure 3B than Figure 2C. Given that Figure 2 is generated from the full human proteome, this suggests that iRefIndex IDed many predictive complexes, and should not be weighted equally with the other three. Similarly, the overlap between databases other than iRefIndex in Figure 2 is much higher in 2C than with Figure 3B. It would be more instructive and easier to visualize if Figure 3B and 2C were both shown as Venn diagrams.

-Based on figure 2C, there are approx. 17,828 PINs that overlap in more than 2 non-iRefIndex databases. This number is 5,610 for the thyroid cell dataset. This number is lower than the 6665 quoted in the text, but is worth considering as it could rule out interactions with less supporting experimental data

Reviewer #2

(Remarks to the Author)

The authors present a robotics-assisted method for sample preparation in Co-Fractionation Mass Spectrometry (CF-MS) experiments. While CF-MS is a powerful technique for studying stable protein interactions, sample preparation for a single experiment typically requires 3-5 days of manual work. The authors use automated laboratory equipment to replace manual performance of two key steps: 1) enzymatic digestion of fractions and 2) peptide cleanup. This roboticization allows more plates to be processed than is feasible manually (performing complex digestion for six plates for 18 hours continuously, and

peptide cleanup for 12 hours continuously). The robotics implementation for sample preparation addresses a genuine bottleneck in CF-MS workflows.

However, my concerns are will the computational analysis. While the addition of robotics to the sample preparation workflow is useful, the data analysis and prediction of protein interactions uses existing tools (CCProfiler and PrInCE). Additionally, I have serious concerns with the construction of the gold standard training set.

Gold standard dataset construction: I'm concerned about the "multi-database search strategy" for constructing training sets. Firstly, it is not so novel to pull examples of known interactions from multiple databases in order to construct a larger training set. It is fairly standard practice gather examples of known interactions from multiple databases in order to construct a larger training set.

While the authors pull protein interactions from four databases (CORUM, Complex Portal, hu.MAP, and iRefIndex), iRefIndex is a large outlier in terms of size and overlap with other sources. Of the ~170,000 interactions in iRefIndex, only 12, 28, and 200 overlap with the other databases. This strongly suggests that the interactions in iRefIndex are not reflective of stable protein interactions that CF-MS is designed to capture and should be removed from the training set. While the iRefIndex resource website is currently down, it does not make sense to me that an interaction database that says it includes CORUM would only contain 28 PPIs in common with CORUM. Additionally, the size of complexes in iRefIndex appears biologically unrealistic (too large) and distinct from curated protein complex databases (Figure 3A). I

This strongly suggests to me that the interactions in iRefIndex are not reflective of stable protein interactions, and should be removed from the training set. Additionally, while the iRefIndex resource website is currently down, it does not make sense to me that an interaction database that says it includes CORUM would only contain 28 PPIs in common with CORUM. Additionally, the size of complexes in the iRefIndex are biologically unrealistic (too large), and a distinct outlier from the curated protein complex databases (Figure 3A). In the refinement stage, 1754 proteins are removed from iRefIndex, which only removes 3 protein complexes.

Interaction labeling approach: If protein pair was labeled as a PPI in any of the four databases, it is labeled as true. This extremely permissive criterion likely incorporates hundreds of thousands of false positives from iRefIndex

Computational analysis: While the statistical techniques employed are appropriate in principle, CCProfiler performs relatively similarly with CORUM, hu.MAP and Complex Portal, but iRefIndex is again a large outlier (Figure 3D). I'm highly concerned about the effect of including ~170,000 protein interactions from iRefIndex as TRUE labels, specially on the calibration of the false discovery rates.

There is a quite low level of overlap between the predicted interactions from CCProfiler and PrInCE. Additionally, 1.4 million co-eluted protein pairs identified by PrInCE (above 20% FDR?) is unrealistically and extremely high. I would like to see a precision-recall curve for the predicted interactions, to understand how the predicted interactions overlap with the curated protein complex databases.

The use of isotonic regression is an appropriate approach to transform raw prediction scores to probability estimates, and linear interpolation is a reasonable approach to estimate precision from scores.

Complex identification: It's not clear to me if the complexes presented in Figures 5 and 6 are the only complexes identified, as this is a very low number that does not seem to include major abundant protein complexes. Compared to other protein complex identification papers from CF-MS, this is a very low number of identified complexes.

Complex modelling: Alphafold predicted protein complex images for PFAS/HK1/TGM2 are not clearly presented, and it is difficult to see where the predicted interfaces are, as both subunits are the same colors. Is it possible to combine these three individual alphafold predictions into a single prediction of the 3-protein complex? If there are major steric clashes between the pairwise subunit predictions when overlapped, it would decrease confidence that this is a true 3-protein complex. These candidate novel interactions could be supported with use of the ipTM score or the IPSAE score (PMID: 39990437).

In Figure 6B, PFAS/HK1/TGM2 are shown to have an extra peak in Nthy around 18m, however, this peak is not visible in the heatmap in figure 6A.

Recommendations:

- Remove iRefIndex from the training set
- Provide a precision-recall curve for the predicted interactions, showing overlap with the curated protein complex databases
- Clarify the number of interactions and complexes identified
- Remove references to multi-database search strategy, as it is standard practice to use multiple databases to construct a gold-standard dataset of known interactions
- Improve presentation of alphafold predicted protein complex images
- Emphasize novelties in the computational strategy compared to past CF-MS papers.

While the robotics approach to sample preparation is innovative and does shorten the experimental time, the computational analysis relies on existing tools and employs a questionable strategy for defining true interactions. The major issue is the use of a large number of PPIs from the iRefIndex database, which is an extreme outlier from other protein complex

databases. With improvements to the computational methodology and more rigorous validation of predictions, this work could make a more substantial contribution to the field.

Reviewer #3

(Remarks to the Author)

Lyu et al. present ProteoAutoNet, a workflow combining robotics-assisted CF-MS sample processing and a multi-database search strategy. The workflow is applied to three thyroid cell lines to identify cancer specific protein-protein interactions. While the automated workflow provides clear benefits with respect to throughput and experimental reproducibility, the computational part of ProteoAutoNet is insufficiently described and benchmarked.

Major points:

- 1) The computational method of ProteoAutoNet for integrating multiple database search results is only very briefly and superficially described in the methods section. The authors should provide more detailed information and a performance benchmark. It is not clear how ProteoAutoNet relates to the findings presented in Figure 3.
- 2) If protein pairs rather than protein complexes are of interest, also alternative strategies, such as SECAT or PCprophet could be evaluated.
- 3) Conceptually, I would generally advise to combine databases prior to CCprofiler or PrInCE analysis (rather than post-processing combination), which will keep statistical models of these tools intact. Have the authors tested this approach?
- 4) While the reported results in the cancer cell lines are certainly interesting, it is not clear whether these were only possible because of the adjusted analysis approach in ProteoAutoNet or whether a traditional CCprofiler or PrInCE analysis would also report these findings.

Minor points:

- 5) Line 61 on page 2: check sentence structure “the performance of the in”

Version 1:

Reviewer comments:

Reviewer #1

(Remarks to the Author)

Authors have addressed the main concerns, the revised paper is now suitable for publication.

Reviewer #4

(Remarks to the Author)

I was asked to evaluate the responses to Reviewers 2 and 3. These appear to be very thorough and well-thought-out responses. I also looked in detail at a few random claims made by the authors and these also checked out

Point by point response to the reviewer's comments:

Reviewer #1 (Remarks to the Author):

Review of NCOMMS-25-11440, "ProteoAutoNet: high-throughput co-eluted protein analysis with robotics and a multi-database search strategy" by Guo and colleagues.

This is an interesting automation workflow with potential broad applications for CFMS style interactomic studies. Overall, the ProteoAutoNet offers the potential as a method for reducing intrasample variability and overall analysis time. However, there are some performance gaps warranting a more thorough comparison between robotically and manually-prepared samples would help to elucidate the relative advantages, and whether there are drawbacks in terms of coverage and replicability.

Reply: We appreciate the reviewer's kind recognition of our work. We have added manually prepared samples and used the coefficient of variation (CV) of protein counts and abundance as benchmarks. The results showed superior protein coverage and quantitative reproducibility in the robotics-assisted platform. Please see our point-by-point response below for the detailed results.

Additionally, integration of the software component isn't fully convincing, as iRefIndex, while well benchmarked, has low overlap with other public databases. Notably, there is approximately a 20% drop in IDs when only considering non-iRefIndex entries, which is worth exploring to improve confidence in the computational aspects of the pipeline.

Reply: Thanks for your suggestions. We have removed iRefIndex based on your and other reviewers' comments. We have employed XGBoost and data augmentation, and used AURPC and AUROC to evaluate the computational confidence. The detailed results are presented in our point-by-point response.

1. Use of 9 unfractionated samples: protein abundance is used as QC measure, but overlap of protein IDs and coverage between replicates as well as comparison to non-robotically prepped samples would provide more objective QC measures

Reply: We appreciate the reviewer's insightful suggestion regarding more objective quality control measures. We have added new experiments that include a non-robotic group to compare protein identification and intensity with the robotic platform. The added text reads as follows:

"We compared the robotics-assisted platform to manual processing using unfractionated TPC-1 samples to validate the performance of the platform further. In the robotical processing group versus the manual processing group, the coefficient of variation (CV) of protein counts decreased from 1.08% to 1.01% (Fig. 1E). The number of proteins confirmed the reliability of the robotics-assisted platform (sFig. 1B). The mean protein counts were comparable robotic (3408 ± 34) and manual processing (3404 ± 37). Protein counts in the robotical group showed 25th and 75th percentiles at 3398 and 3428 respectively, compared to 3381 and 3437 for manual processing. We also evaluated reproducibility by calculating the CV of protein abundance. The results showed superior quantitative reproducibility in the robotic processing group, with a lower median coefficient of variation (20.1% versus 21.6% in manual processing) and a narrower interquartile range (12.6% versus 14.4%) (Fig. 1F)."

We have added the details of the new experiment in "Automated desalting and digestion of co-fractionated proteins" of the Materials and Methods sections:

“To validate the performance of the automated platform, we compared the robotics-assisted processing against manual processing using eight unfractionated TPC-1 samples for each group. Both sets of samples underwent identical processing conditions following the above protocol for denaturation, reduction, alkylation, digestion, and desalting steps.”

2. In general, comparison of robotically-prepped samples to standard workflow would give more robust evidence of benefits of robotic workflow outside of time/effort savings

-Page 4, line 151, authors state that they use 81 raw LC-DIA/MS files to generate their database. A further explanation of these files, species, fractionation, quantity etc. would help elucidate this database. The methods page lists 576 DIA files.

Reply: Thanks for your suggestions. We have added new experiments, which are a non-robotic group to compare protein identification and intensity with the robotic platform as mentioned above. The results showed superior protein coverage and quantitative reproducibility in the robotics-assisted platform.

The 81 raw LC-DIA/MS files were from a public dataset, while the 576 DIA files were generated by us in this study.

The 81 files were obtained from published HEK293 datasets (PMID: 32690956), with fractions collected under optimized conditions as follows: elution range of 10-28 min (0.19 min/fraction), flow rate of 0.5 mL/min, yielding 81 total fractions for quantification. However, to enable a more predictive and generalized framework for identifying co-eluted patterns, we have removed it from the revised manuscript and built an XGBoost model in its place to predict co-eluted proteins, benchmarking its performance with AUPRC and AUROC.

We added the composition of 576 files in “DIA-MS analysis of the cell lines” of Materials and Methods session:

“We acquired a total of 576 analyzable DIA files, including 540 co-fractionated samples from three cell lines with three SEC replicates (60 SEC fractions per replicate collected every 19 seconds between 9-28 minutes), 27 unfractionated quality control samples of automated platform (three technical replicates per SEC replicate), and 9 batch control injections of pooled three cell line mixtures for MS acquisition.”

3. iRefIndex identified many more PINs in Figure 3B than Figure 2C. Given that Figure 2 is generated from the full human proteome, this suggests that iRefIndex IDed many predictive complexes, and should not be weighted equally with the other three. Similarly, the overlap between databases other than iRefIndex in Figure 2 is much higher in 2C than with Figure 3B. It would be more instructive and easier to visualize if Figure 3B and 2C were both shown as Venn diagrams.

Reply: Thank you very much for your suggestions. As you pointed out, the iRefIndex database may introduce substantial noise. Therefore, we have removed iRefIndex and merged the remaining three databases according to your suggestion. In the revised version, to establish a more predictive and generalized framework for identifying co-eluted patterns, we have developed an XGBoost model for co-eluted proteins.

The brief workflow of the XGBoost modeling is presented in the “A machine learning model for protein interaction prediction” of Results session:

“We built an XGBoost model for co-eluted protein prediction and evaluated its performance using the area under the receiver operating characteristic curve (AUROC) and the area under the precision-recall curve (AUPRC).

In the initial phase, the TPC-1 cell line contained 27,794 positive interactions. The remaining interactions were annotated as either negative (6,139,613) or unlabeled (8,548,311). FTC238 showed comparable positive interactions at 27,615 but exhibited 6,255,162 negative and 8,723,199 unlabeled interactions. Notably, Nthy-ori 3-1 displayed 31,076 positive, 8,397,238 negative and 12,926,198 unlabeled interactions. The greater interaction counts in Nthy-ori 3-1 directly reflect its larger proteome coverage (N), since the potential number of pairs scales with $N^2/2$ (sFig. 2A). We evaluated different downsampling ratios and determined that a 1:5 negative-to-positive ratio (precision = 0.28, recall = 0.73) offered the optimal balance, minimizing the excessive false negatives observed at higher ratios. The 1:5 ratio achieved stable precision in random forest models, as confirmed by five-fold cross-validation (sFig. 2B). Density plots verified effective enrichment of high-confidence interactions (PrInCE score > 0.5) under this ratio (sFig. 2C). We selected 20,635 co-eluted proteins with positive labels from TPC-1, 20,373 from FTC238 and 22,947 from Nthy-ori 3-1.

To address the scarcity and limited diversity of known protein-protein interactions (PPIs), we employed data augmentation by introducing random perturbations to positive samples. The second phase involved data augmentation through random value perturbation and missing-value perturbation as described previously (sFig. 2D). The XGBoost model was trained using five key features on group-stratified splits of 44.7 million co-elution traces from three cell lines: TPC-1 with 14.7 million traces (14,715,718), FTC238 with 15.0 million (15,005,976), and Nthy-ori 3-1 with 14.9 million (14,969,894). The final 1:1 balanced dataset was divided into training, validation and test sets, with additional FTC133 data for external validation.”

Additionally, your suggestion to present Figures 3B and 2C as Venn diagrams is well taken. Since we have established a machine learning model in the revised version and no longer need to compare the analysis results with existing software (CCprofler and PrInCE). Venn diagrams are not included in the updated version.

4. Based on figure 2C, there are approx. 17,828 PINs that overlap in more than 2 non-iRefIndex databases. This number is 5,610 for the thyroid cell dataset. This number is lower than the 6665 quoted in the text, but is worth considering as it could rule out interactions with less supporting experimental data

Reply: We thank the reviewer for highlighting this important distinction. The lower number of overlapping PINs in the thyroid cell dataset (5610 vs. 17,828 in non-iRefIndex databases) likely stems from two key factors in our initial analysis:

- 1) The limited predictive power of the original PrInCE (naive Bayes-based model) for identifying co-eluted proteins.
- 2) Overly stringent data/cell strategy correction applied to the data analysis of the three cell lines.

To address these limitations, we have implemented significant improvements in the revised manuscript:

- 1) Combined three complex databases (no iRefIndex) as gold standard.
- 2) Developed a more robust XGBoost-based model to enhance co-elution prediction accuracy.

We described the machine learning framework in “Machine learning framework for predicting PPIs” of Materials and Methods:

“We established a machine learning-based framework for predicting protein-protein interactions by integrating three reference databases as the gold standard and employing XGBoost algorithms. We extracted the column names of databases using R (v4.3.0) and refined using the human FASTA (May 2024) to remove non-matching proteins and complexes, with the counts detailed in Fig. 2A. Complexes from CORUM, Complex Portal and hu.MAP were combined as a human interaction reference dataset. Protein interactions were derived from the set of identified proteins (N), yielding a total of $N \times (N - 1) / 2$ possible interactions. However, the set of known true interactions from gold-standard databases is sparse relative to this theoretical maximum.

We implemented a two-phase strategy to build a robust predictive model from these sparse annotations. In the initial phase, sensitivity-optimized downsampling (1:5 negative-to-positive ratio) was performed across all three cell lines (TPC-1, FTC238, and Nthy-ori 3-1) using the PrInCE R package (v1.16.0, R4.3.0). Protein interactions were annotated against integrated database from CORUM, Complex Portal and hu.MAP, with random forest classification (5-fold CV) identifying high-confidence interactions (score > 0.5) for data augmentation. The second phase involved two-track data augmentation of high-confidence interactions: (1) random value perturbation, where protein elution profile values were scaled by a random factor between 90% and 110%; and (2) missing-value perturbation, where the positions of missing values were randomly either retained in place or shifted to an adjacent time point. This augmentation process was implemented in Python 3.9.21. Representative augmented pairs were selected based on a composite score: $score = (n_valid \times 0.5) + (|Pearson_r| \times 30) - (euc_dist \times 0.01)$, from which the top 40 and bottom 10 scoring pairs were retained. Final training sets combined original and augmented positives and matched negatives (1:1 ratio). The XGBoost classifier (Python 3.9.21) was trained using stratified partitioning via GroupShuffleSplit by protein ID to maintain group integrity (80% training, 10% validation, and 10% test sets). Five interaction features were employed: smoothed and raw Pearson correlation coefficients, Euclidean distance, co-peak frequency, and weighted cross-correlation (WCC) scores. Hyperparameter optimization was conducted through three-fold stratified grid search covering max_depth (4-8), learning_rate (0.01-0.1), subsample ratio (0.6-1.0), with early stopping after 20 rounds of no improvement. Class imbalance was addressed by setting scale_pos_weight to the negative-to-positive ratio. The final model was selected based on AUPRC performance during cross-validation, with optimal parameters retrained on the complete 80% training dataset. Final model performance was evaluated on 10% internal test set and an independent external validation set of FTC133 cell line.

We employed two methods for precision calculation. The first method treated all unknown samples as negative instances (0), and precision was calculated as $TP / (TP + FP)$, where TP denotes true positives and FP denotes false positives. However, this approach may underestimate the true precision, since unknown samples are likely to contain some true positives. Therefore, we also applied a weighted precision method, which assigns the predicted probability of each unknown sample as its positive-class weight. These weighted precisions were combined with labeled positive and negative samples, and all interactions were ranked by confidence to compute a cumulative precision curve.”

We also summarized the benchmarking and performance of the XGBoost model in the Results:

“Model evaluation showed robust internal performance with cross-validation AUROC of 0.81 (Fig. 3A) and AUPRC of 0.82 (Fig. 3B). Hyperparameter optimization in the internal validation set yielded an AUROC of 0.77 (Fig. 3C) and an AUPRC of 0.78 (Fig. 3D). Hold-out test results achieved an AUROC of 0.78 (Fig. 3E) and an AUPRC of 0.79 (Fig. 3F). External validation on the FTC133 dataset at the native sample ratio yielded an AUROC of 0.68 (sFig. 2G). The reduced AUROC during external validation reflects both technical differences across cell lines and the inherent challenge of class imbalance in native proteomic data. For high-confidence predictions using a strict probability threshold of 0.95, weighted precision achieved 0.5 in the FTC133 dataset (Fig. 3G), which contained 6558 labeled interactions out of 18,811 interactions (precision > 0.5) from integrated databases (Fig. 3H). The three-cell consensus set of 25,173 interactions exhibited comparable performance with a probability higher than 0.95 and a weighted precision over 0.6 (sFig. 2E), including 1197 interactions that were previously documented in CORUM (sFig. 2F).”

The above systematic improvements have significantly enhanced the reliability of the model, providing a robust computational framework for PPI study.

Reviewer #2 (Remarks to the Author):

The authors present a robotics-assisted method for sample preparation in Co-Fractionation Mass Spectrometry (CF-MS) experiments. While CF-MS is a powerful technique for studying stable protein interactions, sample preparation for a single experiment typically requires 3-5 days of manual work. The authors use automated laboratory equipment to replace manual performance of two key steps: 1) enzymatic digestion of fractions and 2) peptide cleanup. This roboticization allows more plates to be processed than is feasible manually (performing complex digestion for six plates for 18 hours continuously, and peptide cleanup for 12 hours continuously). The robotics implementation for sample preparation addresses a genuine bottleneck in CF-MS workflows.

However, my concerns are will the computational analysis. While the addition of robotics to the sample preparation workflow is useful, the data analysis and prediction of protein interactions uses existing tools (CCprofiler and PrInCE). Additionally, I have serious concerns with the construction of the gold standard training set.

Gold standard dataset construction: I'm concerned about the "multi-database search strategy" for constructing training sets. Firstly, it is not so novel to pull examples of known interactions from multiple databases in order to construct a larger training set. It is fairly standard practice gather examples of known interactions from multiple databases in order to construct a larger training set.

Reply: Thank you very much for your positive feedback on the ProteoAutoNet automated workflow. We agree that integrating databases is not rare in the literature. Our improved approach introduces several key methodological advancements that enhance the quality and reliability of the machine-learning model for PPI prediction:

We developed an XGBoost model to predict co-eluted proteins using a two-stage sample augmentation strategy to address the scarcity and limited diversity of known true PPIs and enhance the model's ability to generalize. We employed a XGBoost-based model utilizing a two-stage sample augmentation strategy. This approach resulted in a 21.5% improvement in predictive performance on a single dataset, achieving an AUROC of 0.78 in an internal hold-out test dataset. The five inputs of the model as follows:

- a. Pearson correlation of raw elution traces
- b. Pearson correlation of smoothed elution traces
- c. Euclidean distance
- d. Co-peak detection
- e. Weighted cross-correlation (WCC)

The first four inputs were calculated using Python implementations consistent with the PrInCE package (PMID: 33471077), while the WCC was derived from a Python adaptation of the wccsom R package's algorithm (PMID: 22939629).

In summary, we have removed all “multi-database search strategy” from manuscript, including in the title. The revised manuscript, now entitled “**ProteoAutoNet: high-throughput co-eluted protein analysis with robotics and machine learning**” and the revised version not only strengthens the reliability of the gold standard but also enhances the performance of the model, providing an effective computational framework for PPI studies.

While the authors pull protein interactions from four databases (CORUM, Complex Portal,

hu.MAP, and iRefIndex), iRefIndex is a large outlier in terms of size and overlap with other sources. Of the ~170,000 interactions in iRefIndex, only 12, 28, and 200 overlap with the other databases. This strongly suggests that the interactions in iRefIndex are not reflective of stable protein interactions that CF-MS is designed to capture and should be removed from the training set. While the iRefIndex resource website is currently down, it does not make sense to me that an interaction database that says it includes CORUM would only contain 28 PPIs in common with CORUM. Additionally, the size of complexes in iRefIndex appears biologically unrealistic (too large) and distinct from curated protein complex databases (Figure 3A).

This strongly suggests to me that the interactions in iRefIndex are not reflective of stable protein interactions, and should be removed from the training set. Additionally, while the iRefIndex resource website is currently down, it does not make sense to me that an interaction database that says it includes CORUM would only contain 28 PPIs in common with CORUM. Additionally, the size of complexes in the iRefIndex are biologically unrealistic (too large), and a distinct outlier from the curated protein complex databases (Figure 3A). In the refinement stage, 1754 proteins are removed from iRefIndex, which only removes 3 protein complexes.

Interaction labeling approach: If protein pair was labeled as a PPI in any of the four databases, it is labeled as true. This extremely permissive criterion likely incorporates hundreds of thousands of false positives from iRefIndex.

Reply: We sincerely appreciate the reviewer's critical observations regarding iRefIndex. Based on your suggestions and our additional investigations, we have completely excluded iRefIndex from our revised analysis.

We agree that its inclusion introduced biologically unrealistic interactions and compromised the reliability of our gold standard. To directly address your concerns, we re-analyzed the data using only the three curated databases (CORUM, hu.MAP, and Complex Portal). This refinement substantially improved the biological relevance. A key example is the novel PIN: HK-TGM2-PFAS.

In previous analysis using iRefIndex, the interaction between PFAS and HK1 is labeled as true. It was likely spurious, originating from the iRefIndex entry "iRef_4137891". This spurious link resulted in the prediction of a PIN: TGM2-HK1-PFAS. After removing iRefIndex, the PFAS-HK1 link was eliminated. The PFAS-HK1 interaction was predicted with high probability (0.936) in our XGBoost model, but it did not meet our stringent confidence threshold (prob > 0.95).

Consequently, we now more accurately report novel interactions between TGM2 and HK1 (ipTM = 0.61, pTM = 0.66), whereas the removed PFAS-HK1 (ipTM = 0.25, pTM = 0.65) had low confidence in Alphafold3.

This case exhibits that excluding iRefIndex effectively filtered out lower-quality interactions, leading to a more reliable biological relevance of PPIs. We have updated all the figures and results in the manuscript.

Computational analysis: While the statistical techniques employed are appropriate in principle, CCProfiler performs relatively similarly with CORUM, hu.MAP and Complex Portal, but iRefIndex is again a large outlier (Figure 3D). I'm highly concerned about the effect of including ~170,000 protein interactions from iRefIndex as TRUE labels, especially on the calibration of the false discovery rates.

There is a quite low level of overlap between the predicted interactions from CCProfiler and PrInCE. Additionally, 1.4 million co-eluted protein pairs identified by PrInCE (above 20% FDR?) is unrealistically and extremely high. I would like to see a precision-recall curve for the predicted interactions, to understand how the predicted interactions overlap with the curated protein complex databases.

The use of isotonic regression is an appropriate approach to transform raw prediction scores to probability estimates, and linear interpolation is a reasonable approach to estimate precision from scores.

Reply: We sincerely appreciate the reviewer's thorough analysis and valid concerns regarding database consistency and prediction reliability. We agree that the inclusion of iRefIndex and limited diversity of CF-DIA-MS data caused challenges in predicting reliable PPIs.

To address these limitations, we have implemented significant improvements in the revised manuscript:

- 1) Combined three complex databases (no iRefIndex) as gold standard.
- 2) Developed a more robust XGBoost-based model enhanced with data augmentation.

By introducing random perturbations to the known true PPIs, this strategy effectively rebalanced the dataset and mitigated the scarcity and limited diversity of positive samples, thereby enhancing co-elution prediction accuracy. We described the machine learning framework in “Machine learning framework for predicting PPIs” of Materials and Methods:

“We established a machine learning-based framework for predicting protein-protein interactions by integrating three reference databases as the gold standard and employing XGBoost algorithms. We extracted the column names of databases using R (v4.3.0) and refined using the human FASTA (May 2024) to remove non-matching proteins and complexes, with the counts detailed in Fig. 2A. Complexes from CORUM, Complex Portal and hu.MAP were combined as a human interaction reference dataset. Protein interactions were derived from the set of identified proteins (N), yielding a total of $N \times (N - 1) / 2$ possible interactions. However, the set of known true interactions from gold-standard databases is sparse relative to this theoretical maximum.

We implemented a two-phase strategy to build a robust predictive model from these sparse annotations. In the initial phase, sensitivity-optimized downsampling (1:5 negative-to-positive ratio) was performed across all three cell lines (TPC-1, FTC238, and Nthy-ori 3-1) using the PrInCE R package (v1.16.0, R4.3.0). Protein interactions were annotated against integrated database from CORUM, Complex Portal and hu.MAP, with random forest classification (5-fold CV) identifying high-confidence interactions (score > 0.5) for data augmentation. The second phase involved two-track data augmentation of high-confidence interactions: (1) random value perturbation, where protein elution profile values were scaled by a random factor between 90% and 110%; and (2) missing-value perturbation, where the positions of missing values were randomly either retained in place or shifted to an adjacent time point. This augmentation process was implemented in Python 3.9.21. Representative augmented pairs were selected based on a composite score: $score = (n_valid \times 0.5) + (|Pearson_r| \times 30) - (euc_dist \times 0.01)$, from which the top 40 and bottom 10 scoring pairs were retained. Final training sets combined original and augmented positives and matched negatives (1:1 ratio). The XGBoost classifier (Python 3.9.21) was trained using stratified partitioning via GroupShuffleSplit by protein ID to maintain group

integrity (80% training, 10% validation, and 10% test sets). Five interaction features were employed: smoothed and raw Pearson correlation coefficients, Euclidean distance, co-peak frequency, and weighted cross-correlation (WCC) scores. Hyperparameter optimization was conducted through three-fold stratified grid search covering max_depth (4-8), learning_rate (0.01-0.1), subsample_ratio (0.6-1.0), with early stopping after 20 rounds of no improvement. Class imbalance was addressed by setting scale_pos_weight to the negative-to-positive ratio. The final model was selected based on AUPRC performance during cross-validation, with optimal parameters retrained on the complete 80% training dataset. Final model performance was evaluated on 10% internal test set and an independent external validation set of FTC133 cell line.

We employed two methods for precision calculation. The first method treated all unknown samples as negative instances (0), and precision was calculated as $TP / (TP + FP)$, where TP denotes true positives and FP denotes false positives. However, this approach may underestimate the true precision, since unknown samples are likely to contain some true positives. Therefore, we also applied a weighted precision method, which assigns the predicted probability of each unknown sample as its positive-class weight. These weighted precisions were combined with labeled positive and negative samples, and all interactions were ranked by confidence to compute a cumulative precision curve.”

We also summarized the benchmarking and performance of the XGBoost model in the Results:

“Model evaluation showed robust internal performance with cross-validation AUROC of 0.81 (Fig. 3A) and AUPRC of 0.82 (Fig. 3B). Hyperparameter optimization in the internal validation set yielded an AUROC of 0.77 (Fig. 3C) and an AUPRC of 0.78 (Fig. 3D). Hold-out test results achieved an AUROC of 0.78 (Fig. 3E) and an AUPRC of 0.79 (Fig. 3F). External validation on the FTC133 dataset at the native sample ratio yielded an AUROC of 0.68 (sFig. 2G). The reduced AUROC during external validation reflects both technical differences across cell lines and the inherent challenge of class imbalance in native proteomic data. For high-confidence predictions using a strict probability threshold of 0.95, weighted precision achieved 0.5 in the FTC133 dataset (Fig. 3G), which contained 6558 labeled interactions out of 18,811 interactions (precision > 0.5) from integrated databases (Fig. 3H). The three-cell consensus set of 25,173 interactions exhibited comparable performance with a probability higher than 0.95 and a weighted precision over 0.6 (sFig. 2E), including 1197 interactions that were previously documented in CORUM (sFig. 2F).”

Complex identification: It's not clear to me if the complexes presented in Figures 5 and 6 are the only complexes identified, as this is a very low number that does not seem to include major abundant protein complexes. Compared to other protein complex identification papers from CF-MS, this is a very low number of identified complexes.

Reply: We appreciate the reviewer's careful evaluation of our complex identification results. We would like to clarify the key points regarding Figures 5-6 and the overall complex detection:

- 1) In the previous manuscript, Figure 5 and Figure 6 specifically highlight differentially expressed protein interactions between FTC238 vs Nthy-ori 3-1 (Figure 5) and TPC-1 vs Nthy-ori 3-1 (Figure 6), rather than representing the complete complexes.
- 2) The relatively low number of Figure 5 and Figure 6 is caused by employing the criteria in the previous manuscript:

“To enhance the reliability of identified interacting proteins, we established a linear relationship

between retention time and molecular weight using a standard protein mix. Proteins with molecular weights smaller than those listed on UniProt were excluded. Only protein pairs found in at least five biological replicates across two cell lines were retained.”

3) Figure 4A presents all interactions identified at 20% FDR across three cell lines. The relatively lower number compared to other CF-MS studies might stem from the fact that we used the default settings of PrInCE, which employs a Naive Bayes classifier by default, resulting in suboptimal prediction performance.

We totally agree that the previous method identified a limited number of complexes. Therefore, we constructed an XGBoost-based machine learning model to predict PPIs:

We identified over 20,000 interactions using the XGBoost model across three thyroid cell lines, as described in the “A machine learning model for protein interaction prediction” of the Results:

“The three-cell consensus set of 25,173 interactions exhibited comparable performance with a probability higher than 0.95 and a weighted precision over 0.6 (sFig. 2E), including 1197 interactions that were previously documented in CORUM (sFig. 2F).”

The analysis of major complexes was conducted via the following approach.

First, we enriched the biological functions from all interactions across the three cell lines. Our analysis specifically highlighted the top ten KEGG pathways. The added text in “Protein interaction landscape of thyroid cell lines” of the Results as follows:

“The top ten significant KEGG pathways included ribosome, proteasome, biosynthesis of amino acids, DNA replication and cysteine and methionine metabolism (Fig. 4A). Ribosome and proteasome complexes as the top two pathways align with mass spectrometry-based PPI studies (PMID: 33961781, 26186194), further validating the reliability of ProteoAutoNet workflow. We prioritized pathways by combining the p-value and enrichment score. This approach better captures stable protein complexes, leading to focus on five significant protein interaction networks (PINs). The proteasome pathway ranked highest and followed by sulfur metabolism, DNA replication, ribosome, and 2-oxocarboxylic acid metabolism.

ProteoAutoNet detected 37 out of 43 known proteasome components from the KEGG database (sFig. 3A), exhibiting the coverage of 20S core particle with its PSMA and PSMB families (PMID: 12083007), and the 19S regulatory particle with its PSMC and PSMD families (PMID: 19145068). We also detected partial components of regulatory factors including the PSME and PSMF families. The detected 20S-19S-regulatory factor architecture matches the known proteasome organization (PMID: 19145068, 24804812, 30700495). This methodological convergence reinforces the validity of ProteoAutoNet for studying the proteasome complex. The sulfur metabolism (7 out of 10 known components) (sFig. 3B), DNA replication (22/36) (sFig. 3C), ribosome pathway (72/131) (sFig. 3D, E) and 2-oxocarboxylic acid metabolism (10/18) (sFig. 3F) were detected based on KEGG database annotations. We constructed protein interaction networks (PINs) for these pathways and performed Markov clustering (MCL) to identify functional modules.”

Second, we focus on the conserved complexes by the using common interactions across three thyroid cell lines in the “Protein interaction landscape of thyroid cell lines” of the Results:

“We displayed common PINs from three cell lines and highlighted edges with experimental evidence in STRING (sFig. 3G). We identified three functional modules with extensive overlap

with the STRING: translation (**Fig. 4B**), proteasome-ubiquitin (**Fig. 4C**) and protein folding (**Fig. 4D**). As the module with highest overlap of prior knowledge, the detection of 40S/55S/60S ribosomal subunits and eukaryotic initiation factor 3 (eIF3) complex across all three cell lines validates the reliability of ProteoAutoNet results (**Fig. 4B**). Besides, the detection of proteasome corroborates preclinical evidence for proteasome inhibitors in medullary thyroid cancer (MTC) (PMID: 37152939). The Prefoldin and chaperonin-containing TCP-1 (CCT) complexes participate in polypeptide folding, consistent with their canonical roles in eukaryotic protein folding pathways (PMID: 35111762). Some subunits of these complexes have been reported as biomarkers in gastric and lung cancer (PMID: 31957800, 27694898).”

Our revised analysis identifies a comprehensive set of over 20,000 interactions that include major abundant complexes (e.g. ribosome, proteasome), effectively addressing the initial limitation.

Complex modelling: Alphafold predicted protein complex images for PFAS/HK1/TGM2 are not clearly presented, and it is difficult to see where the predicted interfaces are, as both subunits are the same colors. Is it possible to combine these three individual alphafold predictions into a single prediction of the 3-protein complex? If there are major steric clashes between the pairwise subunit predictions when overlapped, it would decrease confidence that this is a true 3-protein complex. These candidate novel interactions could be supported with use of the ipTM score or the IPSAE score (PMID: 39990437).

Reply: We sincerely thank the reviewer for prompting this deeper structural analysis. Following the reviewer’s suggestions on our gold standard, we completely excluded the iRefIndex. This removal led to a re-analysis of our findings. The interaction between PFAS and HK1, which was supported by iRefIndex, was consequently filtered out.

Therefore, we focus on the binary interaction between TGM2 and HK1, which is supported by prob > 0.95 in XGBoost model and weighted precision > 0.6 in thyroid cell lines. It had confident scores from AlphaFold3: ipTM = 0.61, pTM = 0.66.

In direct response to the reviewer’s request, we employed the IPSAE score (PMID: 39990437) to provide a rigorous assessment of TGM2-HK1 interface.

Chn1	Chn2	PAE	Dist	Type	ipSAE	ipSAE_d0chn	ipSAE_d0dom	ipTM_af	ipTM_d0chn	pDockQ	pDockQ2	LIS
A	B		10	10 asym	0.112392	0.665656	0.253077	0.61	0.483943	0.4617	0.0364	0.0658
B	A		10	10 asym	0.013751	0.674476	0.131875	0.61	0.37985	0.4617	0.0294	0.0438
A	B		10	10 max	0.112392	0.674476	0.253077	0.61	0.483943	0.4617	0.0364	0.0548

The structural analysis yields a high confident model quality (ipTM = 0.61, pTM = 0.66) but a low interface-specific score (ipSAE = 0.112). We suppose the discrepancy is not a false negative but a

reflection of the unique structural biology of Hexokinase 1 (HK1). HK1 is a dynamic allosteric enzyme (PMID: 10686099). In the previous structural study, it exists primarily as a monomer in solution and undergoes large ligand-dependent conformational changes and rigid-body rotations between its N and C-terminal halves. High ipTM/pTM scores indicate HK1 and TGM2 are well-folded proteins. However, the low IPSAE score suggests HK1 does not expose the necessary interface for stable binary interaction with TGM2. HK1 might be induced by post-translational modifications or metabolites to form a specific and transient interaction.

We thank the reviewer for prompting these critical improvements of the structural explanation.

In Figure 6B, PFAS/HK1/TGM2 are shown to have an extra peak in Nthy around 18m, however, this peak is not visible in the heatmap in figure 6A.

Reply: We sincerely appreciate the reviewer's insightful observation. In the original submission, the heatmap was generated using data from one representative biological replicate of each cell line, while the line plots displayed the mean values across all three biological replicates. We acknowledge that this inconsistency in data presentation could lead to potential misinterpretation.

In the revised manuscript we used the mean values in heatmap and protein traces:

An additional peak in Nthy-ori 3-1 can be observed around 17 minutes and 14 seconds in the heatmap. It is presented in **sFig. 9 and 10** of the revised manuscript.

Recommendations:

- Remove iRefIndex from the training set

Reply: Thanks for your suggestions. We have removed iRefIndex from the gold standard.

- Provide a precision-recall curve for the predicted interactions, showing overlap with the curated protein complex databases

Reply: Thanks for your suggestions. We have added the AUPRC and AUROC plots (**Fig. 3**) to the revised computational methods section and have included an analysis of the overlap between CORUM and our predicted interactions (**sFig. 2F**). The added text in Results as follows:

“Model evaluation showed robust internal performance with cross-validation AUROC of 0.81 (Fig. 3A) and AUPRC of 0.82 (Fig. 3B). Hyperparameter optimization in the internal validation set yielded an AUROC of 0.77 (Fig. 3C) and an AUPRC of 0.78 (Fig. 3D). Hold-out test results achieved an AUROC of 0.78 (Fig. 3E) and an AUPRC of 0.79 (Fig. 3F). External validation on the FTC133 dataset at the native sample ratio yielded an AUROC of 0.68 (sFig. 2G). The reduced AUROC during external validation reflects both technical differences across cell lines and the inherent challenge of class imbalance in native proteomic data. For high-confidence predictions using a strict probability threshold of 0.95, weighted precision achieved 0.5 in the FTC133 dataset (Fig. 3G), which contained 6558 labeled interactions out of 18,811 interactions (precision > 0.5) from integrated databases (Fig. 3H). The three-cell consensus set of 25,173 interactions exhibited comparable performance with a probability higher than 0.95 and a weighted precision over 0.6 (sFig. 2E), including 1197 interactions that were previously documented in CORUM (sFig. 2F).

Notably, our approach overcomes a critical limitation of existing co-elution proteins prediction that require multi-dataset integration for reliable performance (PMID: 34211188). Previous single-dataset implementations showed limited predictive accuracy with AUROC of 0.65 as reported (PMID: 34211188). Through systematic integration of data augmentation with XGBoost algorithm, we achieved a 21.5% improvement in single-dataset predictive accuracy, while maintaining robust external validation performance at the native class imbalance ratio.”

- Clarify the number of interactions and complexes identified

Reply: Thanks for your suggestions. We have clarified that we identified 25,173 interactions in the Results:

“For high-confidence predictions using a strict probability threshold of 0.95, weighted precision achieved 0.5 in the FTC133 dataset (Fig. 3G), which contained 6558 labeled interactions out of 18,811 interactions (precision > 0.5) from integrated databases (Fig. 3H). The three-cell consensus set of 25,173 interactions exhibited comparable performance with a probability higher than 0.95 and a weighted precision over 0.6 (sFig. 2E), including 1197 interactions that were previously documented in CORUM (sFig. 2F).”

- Remove references to multi-database search strategy, as it is standard practice to use multiple databases to construct a gold-standard dataset of known interactions

Reply: Thanks for your suggestions. We have removed all “multi-database search strategy” in the manuscript. The revised manuscript, now entitled “*ProteoAutoNet: high-throughput co-eluted protein analysis with robotics and machine learning*”

- Improve presentation of alphafold predicted protein complex images

Reply: Thanks for your suggestions. We added X-Y and X-Y-Z perspectives of interactions in main figure (Fig. 5D) The added text reads in “Identification of novel interactions and potential therapeutic targets” as follows:

“The structure of the novel interactions was further supported by AlphaFold 3 (Fig. 5D). Two structural perspectives (X-Y and X-Y-Z) are displayed. The model shows moderate-to-high confidence, with pTM and ipTM scores of 0.66 and 0.61, respectively. It indicates a valuable predicted interaction interface, providing strong structural support for the robustness of this interaction.”

-Emphasize novelties in the computational strategy compared to past CF-MS papers.

Reply: Thanks for your suggestions. In revised manuscript, the computational part of ProteoAutoNet overcomes a critical limitation in current co-elution protein prediction that requires multi-dataset integration for reliable performance (PMID: 34211188).

We have clarified the advances and compared the performance of machine learning model in the Discussion, and we have built the first XGBoost model in PPI studies:

“The computational component of ProteoAutoNet addresses a fundamental challenge in co-elution-based protein prediction: the inherent scarcity and limited diversity of positive training samples⁹. By combining data augmentation with the XGBoost algorithm, we achieved a 21.5% improvement in single-dataset predictive accuracy, attaining an AUROC of 0.78 in internal testing dataset, while maintaining external validation performance (AUROC = 0.68). The XGBoost-based machine learning model in ProteoAutoNet showed consistent predictive accuracy, with precision exceeding 0.6 for 25,173 interactions across three thyroid cell line, and predicted 18,811 protein interactions in the independent FTC133 validation set.”

While the robotics approach to sample preparation is innovative and does shorten the experimental time, the computational analysis relies on existing tools and employs a questionable strategy for defining true interactions. The major issue is the use of a large number of PPIs from the iRefIndex database, which is an extreme outlier from other protein complex databases. With improvements to the computational methodology and more rigorous validation of predictions, this work could make a more substantial contribution to the field.

Reply: We thank the reviewer for the comprehensive assessment of our computational approach. We fully agree that we should remove the iRefIndex and integrated other three databases as gold standard.

As detailed above, we have completely overhauled our computational methodology: (1) removing iRefIndex; (2) combining CORUM, hu.MAP and Complex Portal as gold standard; (3) employing a XGBoost model with data augmentation. With the improvements to our computational methodology, including ipTM from AlphaFold 3 and IPSAE score, we are confident the revised manuscript makes a more reliable contribution to the field.

Reviewer #3 (Remarks to the Author):

Lyu et al. present ProteoAutoNet, a workflow combining robotics-assisted CF-MS sample processing and a multi-database search strategy. The workflow is applied to three thyroid cell lines to identify cancer specific protein-protein interactions. While the automated workflow provides clear benefits with respect to throughput and experimental reproducibility, the computational part of ProteoAutoNet is insufficiently described and benchmarked.

Major points:

1) The computational method of ProteoAutoNet for integrating multiple database search results is only very briefly and superficially described in the methods section. The authors should provide more detailed information and a performance benchmark. It is not clear how ProteoAutoNet relates to the findings presented in Figure 3.

Reply: We sincerely appreciate the reviewer's insightful comments regarding the computational methodology of ProteoAutoNet. Below we provide a comprehensive description of the revised and original computational workflow:

The revised computational part in “Machine learning framework for predicting PPIs” of Materials and Methods is:

“Protein interactions were derived from the set of identified proteins (N), yielding a total of $N \times (N - 1) / 2$ possible interactions. However, the set of known true interactions from gold-standard databases is sparse relative to this theoretical maximum.

We implemented a two-phase strategy to build a robust predictive model from these sparse annotations. In the initial phase, sensitivity-optimized downsampling (1:5 negative-to-positive ratio) was performed across all three cell lines (TPC-1, FTC238, and Nthy-ori 3-1) using the PrInCE R package (v1.16.0, R4.3.0). Protein interactions were annotated against integrated database from CORUM, Complex Portal and hu.MAP, with random forest classification (5-fold CV) identifying high-confidence interactions (score > 0.5) for data augmentation. The second phase involved two-track data augmentation of high-confidence interactions: (1) random value perturbation, where protein elution profile values were scaled by a random factor between 90% and 110%; and (2) missing-value perturbation, where the positions of missing values were randomly either retained in place or shifted to an adjacent time point. This augmentation process was implemented in Python 3.9.21. Representative augmented pairs were selected based on a composite score: $score = (n_valid \times 0.5) + (|Pearson_r| \times 30) - (euc_dist \times 0.01)$, from which the top 40 and bottom 10 scoring pairs were retained. Final training sets combined original and augmented positives and matched negatives (1:1 ratio). The XGBoost classifier (Python 3.9.21) was trained using stratified partitioning via GroupShuffleSplit by protein ID to maintain group integrity (80% training, 10% validation, and 10% test sets). Five interaction features were employed: smoothed and raw Pearson correlation coefficients, Euclidean distance, co-peak frequency, and weighted cross-correlation (WCC) scores. Hyperparameter optimization was conducted through three-fold stratified grid search covering max_depth (4-8), learning_rate (0.01-0.1), subsample_ratio (0.6-1.0), with early stopping after 20 rounds of no improvement. Class imbalance was addressed by setting scale_pos_weight to the negative-to-positive ratio. The final model was selected based on AUPRC performance during cross-validation, with optimal parameters retrained on the complete 80% training dataset. Final model performance was

evaluated on 10% internal test set and an independent external validation set of FTC133 cell line. We employed two methods for precision calculation. The first method treated all unknown samples as negative instances (0), and precision was calculated as $TP / (TP + FP)$, where TP denotes true positives and FP denotes false positives. However, this approach may underestimate the true precision, since unknown samples are likely to contain some true positives. Therefore, we also applied a weighted precision method, which assigns the predicted probability of each unknown sample as its positive-class weight. These weighted precisions were combined with labeled positive and negative samples, and all interactions were ranked by confidence to compute a cumulative precision curve.”

We used the area under the ROC curve (AUROC) and the area under the precision-recall curve (AUPRC) as benchmarks to evaluate the machine learning model in the Results:

“Model evaluation showed robust internal performance with cross-validation AUROC of 0.81 (Fig. 3A) and AUPRC of 0.82 (Fig. 3B). Hyperparameter optimization in the internal validation set yielded an AUROC of 0.77 (Fig. 3C) and an AUPRC of 0.78 (Fig. 3D). Hold-out test results achieved an AUROC of 0.78 (Fig. 3E) and an AUPRC of 0.79 (Fig. 3F). External validation on the FTC133 dataset at the native sample ratio yielded an AUROC of 0.68 (sFig. 2G). The reduced AUROC during external validation reflects both technical differences across cell lines and the inherent challenge of class imbalance in native proteomic data. For high-confidence predictions using a strict probability threshold of 0.95, weighted precision achieved 0.5 in the FTC133 dataset (Fig. 3G), which contained 6558 labeled interactions out of 18,811 interactions (precision > 0.5) from integrated databases (Fig. 3H). The three-cell consensus set of 25,173 interactions exhibited comparable performance with a probability higher than 0.95 and a weighted precision over 0.6 (sFig. 2E), including 1197 interactions that were previously documented in CORUM (sFig. 2F).

Notably, our approach overcomes a critical limitation of existing co-elution proteins prediction that require multi-dataset integration for reliable performance (PMID: 34211188). Previous single-dataset implementations showed limited predictive accuracy with AUROC of 0.65 as reported (PMID: 34211188). Through systematic integration of data augmentation with XGBoost algorithm, we achieved a 21.5% improvement in single-dataset predictive accuracy, while maintaining robust external validation performance at the native class imbalance ratio.”

2) If protein pairs rather than protein complexes are of interest, also alternative strategies, such as SECAT or PCprophet could be evaluated.

Reply: We thank the reviewer for suggesting alternative analytical tools such as SECAT or PCprophet for protein pair analysis. SECAT and PCprophet are indeed powerful methods for identifying protein complexes and tracking their dynamics across conditions. While tools like SECAT and PCprophet excel at complex-level analysis, our goal was to develop a robust machine learning model optimized for pairwise PPI prediction under a single dataset. We evaluated our model's performance against the benchmark set by previous machine learning efforts in the field. The added text is in the “A machine learning model for protein interaction prediction” of Results:

“Notably, our approach overcomes a critical limitation of existing co-elution proteins prediction that require multi-dataset integration for reliable performance (PMID: 34211188). Previous single-dataset implementations showed limited predictive accuracy with AUROC of 0.65 as reported (PMID: 34211188). Through systematic integration of data augmentation with XGBoost algorithm, we achieved a 21.5% improvement in single-dataset predictive accuracy, while maintaining robust

external validation performance at the native class imbalance ratio.”

We believe our computational workflow provides a valuable and complementary approach to the existing toolkit.

3) Conceptually, I would generally advise to combine databases prior to CCprofiler or PrInCE analysis (rather than post-processing combination), which will keep statistical models of these tools intact. Have the authors tested this approach?

Reply: We sincerely appreciate this valuable methodological suggestion.

1) We removed iRefIndex and combined CORUM, huMAP, Complex Portal as gold standard: Our previous benchmarking using a Naive Bayes model in PrInCE R package on the TPC-1 cell line. The precision-recall curve is shown below, yielding an AUROC of 0.026.

We speculated the poor performance was due to both the limitations of the Naive Bayes and the overlooking of the class imbalance in the native dataset.

2) We finally used an improved workflow:

Key enhancements of the new workflow over the previous version included: removal of the iRefIndex database and development of an XGBoost model enhanced with data augmentation. By introducing random perturbations to the known true PPIs, this approach not only rebalanced the positive and negative samples but also effectively addressed their scarcity and limited diversity of positive samples.

The brief workflow of the XGBoost modeling is presented in the “A machine learning model for protein interaction prediction” of Results:

“We built an XGBoost model for co-eluted protein prediction and evaluated its performance using the area under the receiver operating characteristic curve (AUROC) and the area under the precision-recall curve (AUPRC).

In the initial phase, the TPC-1 cell line contained 27,794 positive interactions. The remaining interactions were annotated as either negative (6,139,613) or unlabeled (8,548,311). FTC238 showed comparable positive interactions at 27,615 but exhibited 6,255,162 negative and 8,723,199 unlabeled interactions. Notably, Nthy-ori 3-1 displayed 31,076 positive, 8,397,238 negative and 12,926,198 unlabeled interactions. The greater interaction counts in Nthy-ori 3-1 directly reflect its larger proteome coverage (N), since the potential number of pairs scales with $N^2/2$ (sFig. 2A). We evaluated different downsampling ratios and determined that a 1:5 negative-to-positive ratio (precision = 0.28, recall = 0.73) offered the optimal balance, minimizing the excessive false negatives observed at higher ratios. The 1:5 ratio achieved stable precision in

random forest models, as confirmed by five-fold cross-validation (sFig. 2B). Density plots verified effective enrichment of high-confidence interactions (PrInCE score > 0.5) under this ratio (sFig. 2C). We selected 20,635 co-eluted proteins with positive labels from TPC-1, 20,373 from FTC238 and 22,947 from Nthy-ori 3-1.

To address the scarcity and limited diversity of known protein-protein interactions (PPIs), we employed data augmentation by introducing random perturbations to positive samples. The second phase involved data augmentation through random value perturbation and missing-value perturbation as described previously (sFig. 2D). The XGBoost model was trained using five key features on group-stratified splits of 44.7 million co-elution traces from three cell lines: TPC-1 with 14.7 million traces (14,715,718), FTC238 with 15.0 million (15,005,976), and Nthy-ori 3-1 with 14.9 million (14,969,894). The final 1:1 balanced dataset was divided into training, validation and test sets, with additional FTC133 data for external validation.”

The comprehensive steps of modeling are described in the Materials and Methods:

“Protein interactions were derived from the set of identified proteins (N), yielding a total of $N \times (N - 1) / 2$ possible interactions. However, the set of known true interactions from gold-standard databases is sparse relative to this theoretical maximum.

We implemented a two-phase strategy to build a robust predictive model from these sparse annotations. In the initial phase, sensitivity-optimized downsampling (1:5 negative-to-positive ratio) was performed across all three cell lines (TPC-1, FTC238, and Nthy-ori 3-1) using the PrInCE R package (v1.16.0, R4.3.0). Protein interactions were annotated against integrated database from CORUM, Complex Portal and hu.MAP, with random forest classification (5-fold CV) identifying high-confidence interactions (score > 0.5) for data augmentation. The second phase involved two-track data augmentation of high-confidence interactions: (1) random value perturbation, where protein elution profile values were scaled by a random factor between 90% and 110%; and (2) missing-value perturbation, where the positions of missing values were randomly either retained in place or shifted to an adjacent time point. This augmentation process was implemented in Python 3.9.21. Representative augmented pairs were selected based on a composite score: $score = (n_valid \times 0.5) + (|Pearson_r| \times 30) - (euc_dist \times 0.01)$, from which the top 40 and bottom 10 scoring pairs were retained. Final training sets combined original and augmented positives and matched negatives (1:1 ratio). The XGBoost classifier (Python 3.9.21) was trained using stratified partitioning via GroupShuffleSplit by protein ID to maintain group integrity (80% training, 10% validation, and 10% test sets). Five interaction features were employed: smoothed and raw Pearson correlation coefficients, Euclidean distance, co-peak frequency, and weighted cross-correlation (WCC) scores. Hyperparameter optimization was conducted through three-fold stratified grid search covering `max_depth` (4-8), `learning_rate` (0.01-0.1), `subsample` ratio (0.6-1.0), with early stopping after 20 rounds of no improvement. Class imbalance was addressed by setting `scale_pos_weight` to the negative-to-positive ratio. The final model was selected based on AUPRC performance during cross-validation, with optimal parameters retrained on the complete 80% training dataset. Final model performance was evaluated on 10% internal test set and an independent external validation set of FTC133 cell line

We employed two methods for precision calculation. The first method treated all unknown samples as negative instances (0), and precision was calculated as $TP / (TP + FP)$, where TP denotes true positives and FP denotes false positives. However, this approach may underestimate the true

precision, since unknown samples are likely to contain some true positives. Therefore, we also applied a weighted precision method, which assigns the predicted probability of each unknown sample as its positive-class weight. These weighted precisions were combined with labeled positive and negative samples, and all interactions were ranked by confidence to compute a cumulative precision curve.”

We used the area under the ROC curve (AUROC) and the area under the precision-recall curve (AUPRC) as performance benchmarks to evaluate the machine learning model in the Results:

“Model evaluation showed robust internal performance with cross-validation AUROC of 0.81 (Fig. 3A) and AUPRC of 0.82 (Fig. 3B). Hyperparameter optimization in the internal validation set yielded an AUROC of 0.77 (Fig. 3C) and an AUPRC of 0.78 (Fig. 3D). Hold-out test results achieved an AUROC of 0.78 (Fig. 3E) and an AUPRC of 0.79 (Fig. 3F). External validation on the FTC133 dataset at the native sample ratio yielded an AUROC of 0.68 (sFig. 2G). The reduced AUROC during external validation reflects both technical differences across cell lines and the inherent challenge of class imbalance in native proteomic data. For high-confidence predictions using a strict probability threshold of 0.95, weighted precision achieved 0.5 in the FTC133 dataset (Fig. 3G), which contained 6558 labeled interactions out of 18,811 interactions (precision > 0.5) from integrated databases (Fig. 3H). The three-cell consensus set of 25,173 interactions exhibited comparable performance with a probability higher than 0.95 and a weighted precision over 0.6 (sFig. 2E), including 1197 interactions that were previously documented in CORUM (sFig. 2F).

Notably, our approach overcomes a critical limitation of existing co-elution proteins prediction that require multi-dataset integration for reliable performance (PMID: 34211188). Previous single-dataset implementations showed limited predictive accuracy with AUROC of 0.65 as reported (PMID: 34211188). Through systematic integration of data augmentation with XGBoost algorithm, we achieved a 21.5% improvement in single-dataset predictive accuracy, while maintaining robust external validation performance at the native class imbalance ratio.”

4) While the reported results in the cancer cell lines are certainly interesting, it is not clear whether these were only possible because of the adjusted analysis approach in ProteoAutoNet or whether a traditional CCprofiler or PrInCE analysis would also report these findings.

Reply: We appreciate the reviewer's insightful question regarding the contribution of our improved analytical approach. The identified interactions were uniquely recovered by our ProteoAutoNet workflow and could not have been obtained using a traditional CCprofiler or PrInCE analysis on our dataset. As shown above, the Naive Bayes modules of PrInCE achieved an AUROC of only 0.026, preventing reliable pipeline execution. Therefore, we benchmarked the major complexes from over 20,000 high-confidence interactions against published AP-MS/CF-MS data, as described above.

The added text in “Protein interaction landscape of thyroid cell lines” of the Results as follows:

“The top ten significant KEGG pathways included ribosome, proteasome, biosynthesis of amino acids, DNA replication and cysteine and methionine metabolism (Fig. 4A). Ribosome and proteasome complexes as the top two pathways align with mass spectrometry-based PPI studies (PMID: 33961781, 26186194), further validating the reliability of ProteoAutoNet workflow. We prioritized pathways by combining the p-value and enrichment score. This approach better captures stable protein complexes, leading to focus on five significant protein interaction networks

(PINs). The proteasome pathway ranked highest and followed by sulfur metabolism, DNA replication, ribosome, and 2-oxocarboxylic acid metabolism.

ProteoAutoNet detected 37 out of 43 known proteasome components from the KEGG database (sFig. 3A), exhibiting the coverage of 20S core particle with its PSMA and PSMB families (PMID: 12083007), and the 19S regulatory particle with its PSMC and PSMD families (PMID: 19145068). We also detected partial components of regulatory factors including the PSME and PSMF families. The detected 20S-19S-regulatory factor architecture matches the known proteasome organization (PMID: 19145068, 24804812, 30700495). This methodological convergence reinforces the validity of ProteoAutoNet for studying the proteasome complex. The sulfur metabolism (7 out of 10 known components) (sFig. 3B), DNA replication (22/36) (sFig. 3C), ribosome pathway (72/131) (sFig. 3D, E) and 2-oxocarboxylic acid metabolism (10/18) (sFig. 3F) were detected based on KEGG database annotations. We constructed protein interaction networks (PINs) for these pathways and performed Markov clustering (MCL) to identify functional modules.”

Second, we focus on the conserved complexes by the using common interactions across three thyroid cell lines in the “Protein interaction landscape of thyroid cell lines” of the Results: “We displayed common PINs from three cell lines and highlighted edges with experimental evidence in STRING (sFig. 3G). We identified three functional modules with extensive overlap with the STRING: translation (Fig. 4B), proteasome-ubiquitin (Fig. 4C) and protein folding (Fig. 4D). As the module with highest overlap of prior knowledge, the detection of 40S/55S/60S ribosomal subunits and eukaryotic initiation factor 3 (eIF3) complex across all three cell lines validates the reliability of ProteoAutoNet results (Fig. 4B). Besides, the detection of proteasome corroborates preclinical evidence for proteasome inhibitors in medullary thyroid cancer (MTC) (PMID: 37152939). The Prefoldin and chaperonin-containing TCP-1 (CCT) complexes participate in polypeptide folding, consistent with their canonical roles in eukaryotic protein folding pathways (PMID: 35111762). Some subunits of these complexes have been reported as biomarkers in gastric and lung cancer (PMID: 31957800, 27694898).”

Minor points:

5) Line 61 on page 2: check sentence structure “the performance of the in”

Reply: Thanks for your suggestions, we have revised the manuscript.